# G²N² : Weisfeiler and Lehman go grammatical

**Jason Piquenot**[*]     **Aldo Moscatelli**[*]     **Maxime Bérar**[*]     **Pierre Héroux**[*]

**Jean-Yves Ramel**[†]     **Romain Raveaux**[†]     **Sébastien Adam**[*]

## Abstract

This paper introduces a framework for formally establishing a connection between a portion of an algebraic language and a Graph Neural Network (GNN). The framework leverages Context-Free Grammars (CFG) to organize algebraic operations into generative rules that can be translated into a GNN layer model. As CFGs derived directly from a language tend to contain redundancies in their rules and variables, we present a grammar reduction scheme. By applying this strategy, we define a CFG that conforms to the third-order Weisfeiler-Lehman (3-WL) test using MATLANG. From this 3-WL CFG, we derive a GNN model, named G²N², which is provably 3-WL compliant. Through various experiments, we demonstrate the superior efficiency of G²N² compared to other 3-WL GNNs across numerous downstream tasks. Specifically, one experiment highlights the benefits of grammar reduction within our framework.

## 1 Introduction

In the last few years, the Weisfeiler-Lehman (WL) hierarchy, based on the eponymous polynomial-time isomorphism test (Lehman and Weisfeiler (1968)), has been the most common way to characterise the expressive power of Graph Neural Networks (GNNs) (Morris et al. (2019); Bodnar et al. (2021b;a); Zhang et al. (2023b)). A founding result was the proof that Message Passing Neural Networks (MPNNs) (Gilmer et al. (2017); Wu et al. (2020)) are at most as powerful as the first-order WL test (1-WL) (Morris et al. (2019); Xu et al. (2019)). As a consequence of this result, many subsequent contributions have focused on going beyond this 1-WL limit, to reach more expressive GNNs. For instance, subgraph-based GNNs (Chen et al. (2020); Zhang and Li (2021); Zhao et al. (2022)) succeed to surpass 1-WL expressive power but are still bounded by 3-WL (Frasca et al. (2022)).

One way to ensure $k$-WL expressive power is to mimic one iteration of the $k$-WL test (Maron et al. (2019a)) for each GNN layer. Taking as root the colouring and hashing steps of the $k$-WL algorithm, Maron et al. (2019a) shows that $k$-IGN, based on the basis of equivariant operators defined for IGN (Maron et al. (2019b)), is as powerful as the $k$-WL test. Since $k$-IGN works on $k$-th order tensors and since the cardinal of the basis is equal to the $2k$-th Bell number, it is limited in practice by both the layer input memory consumption and the cardinal of IGN operator basis, even for $k = 3$ (Li et al. (2020)). Concurrently, Provably Powerful Graph Network (PPGN) was also proposed in Maron et al. (2019a). It is able to mimic the second-order Folklore WL test (2-FWL[1]) colouring and hashing steps with MLPs that are coupled together with matrix multiplication. Since PPGN only relies on matrices, it is a more tractable 3-WL architecture than 3-IGN (Zhang et al. (2023a)).

Taking an algebraic point of view, the groundbreaking paper Geerts (2020) reformulates the 1-WL and 3-WL tests as languages based on specific subsets of algebraic operations applied on the adjacency matrix. These fragments of the matrix language MATLANG (Brijder et al. (2019)) called $ML(\mathcal{L}_1)$ and $ML(\mathcal{L}_3)$ are shown to be as expressive as 1-WL and 3-WL (Geerts (2020)). Derived from this result, a model called GNNML1 was proposed in Balcilar

---

[*]LITIS Lab, University of Rouen Normandy, France

[†]LIFAT Lab, University of Tours, France

[1]known to be equivalent to 3-WL test (Huang and Villar (2021))

et al. (2020). GNNML1 is proven to be 1-WL equivalent since it is able to generate any sentence of ML $(\mathcal{L}_1)$. A more expressive model called GNNML3 was proposed in the same paper. It is only shown to be more expressive than 1-WL. This is due to the lack of a systematic procedure of deriving a GNN model from a given language fragment.

In this paper, we leverage this bottleneck by proposing a generic methodology to produce a GNN from any fragment of an algebraic language, opening a new way to ensure expressiveness. The rationale behind our framework is to instantiate a language fragment by a reduced set of generative rules, translated into layer components of a GNN. Starting from the operations set $\mathcal{L}_3$, we build an exhaustive Context-Free Grammar (CFG) able to generate ML $(\mathcal{L}_3)$. This CFG is reduced to remove unnecessary operations among the rules while keeping the equivalence with 3-WL. From the variables of this reduced CFG, GNN inputs are easily deduced. Then, the rules of the CFG determine the GNN layers update functions. As a result of this methodology, we propose a new model called Grammatical Graph Neural Network $(G^2N^2)$ that is provably 3-WL.

The contributions of this work are the following : (i) A generic framework to design a GNN from any fragment of an algebraic language; (ii) The instantiation of the framework on ML $(\mathcal{L}_3)$ resulting in $G^2N^2$, a provably 3-WL GNN; (iii); An experimental validation of the set of rules; (iv) Numerous experiments demonstrating that $G^2N^2$ outperforms existing 3-WL GNNs on various downstream tasks.

The paper is structured as follows. Section 2 introduces the necessary background, by defining MATLANG, its link with WL and CFGs. Section 3 describes our framework and presents the resulting $G^2N^2$ architecture, which is experimentally evaluated in section 4.

## 2 From MATLANG and Weisfeiler-Lehman to Context-Free Grammars and Languages

Let $\mathcal{G} = (\mathcal{V}, \mathcal{E})$ be an undirected graph where $\mathcal{V} = [\![1, n]\!]$ is the set of $n$ nodes and $\mathcal{E} \subset \mathcal{V} \times \mathcal{V}$ is the set of edges. The adjacency matrix $A \in \{0, 1\}^{n \times n}$ represents the connectivity of $\mathcal{G}$.

**Definition 2.1** (MATLANG (Brijder et al. (2019)))
MATLANG is a matrix language with an allowed operation set $\{+, \cdot, \odot, {}^{\mathbf{T}}, \mathrm{Tr}, \mathrm{diag}, \mathbb{1}, \times, f\}$ denoting respectively matrix addition, matrix and element-wise multiplications, transpose and trace computations, diagonal matrix creation from a vector, column vector of 1 generation, scalar multiplication, and element-wise function applied on a scalar, a vector or a matrix. Restricting the set of operations to a subset $\mathcal{L}$ defines a fragment of MATLANG denoted ML $(\mathcal{L})$. $s(X) \in \mathbb{R}$ is a sentence in ML $(\mathcal{L})$ if it consists of consistent consecutive operations in $\mathcal{L}$, operating on a given matrix $X$, resulting in a scalar value. *As an example, $s(X) = \mathbb{1}^{\mathbf{T}} \left( X^2 \odot \mathrm{diag}(\mathbb{1}) \right) \mathbb{1}$ is a sentence of ML $\left( \{\cdot, {}^{\mathbf{T}}, \mathbb{1}, \mathrm{diag}, \odot\} \right)$ computing the trace of $X^2$.*

Equivalences between ML $(\mathcal{L}_1)$ and ML $(\mathcal{L}_3)$ with $\mathcal{L}_1 = \{\cdot, {}^{\mathbf{T}}, \mathbb{1}, \mathrm{diag}\}$, $\mathcal{L}_3 = \{\cdot, {}^{\mathbf{T}}, \mathbb{1}, \mathrm{diag}, \odot\}$ and respectively the 1-WL and 3-WL tests are shown in Geerts (2020): two graphs are indistinguishable by the 1-WL (resp. 3-WL) test if and only if applying any sentence of ML $(\mathcal{L}_1)$ (resp. ML $(\mathcal{L}_3)$) to their adjacency matrices gives the same scalar. Adding $\{+, \times, f\}$ does not improve the expressive power of the fragment (Geerts (2020)).

Transposed in a Machine Learning context, a MATLANG-based GNN will inherit the 3-WL expressive power of ML $(\mathcal{L}_3)$ if it is able to generate any sentence of the fragment while learning the downstream task. To reach this objective, we will instantiate the fragment as a Context Free Language, entirely described by a set of production rules[2].

**Definition 2.2** (Context-Free Grammar and Language)
A Context-Free Grammar (CFG) $G$ is a 4-tuple $(V, \Sigma, R, S)$ with $V$ a finite set of variables, $\Sigma$ a finite set of terminal symbols, $R$ a finite set of rules $V \rightarrow (V \cup \Sigma)^*$, $S$ a start variable. *R completely describes a CFG with the convention that $S$ is placed on the top left.*

---

[2]Figure 7 in appendix A illustrates the process of sentence generation from a grammar.

$B$ is a Context-Free Language (CFL) if there exists a CFG $G$ such that $B = L(G) := \{w, w \in \Sigma^* \text{ and } S \xRightarrow{*} w\}$ where $S \xRightarrow{*} w$ denotes that $S$ can be transformed into $w$ by applying an arbitrary number of rules in $G$.

## 3    FROM ML $(\mathcal{L}_3)$ TO THE 3-WL $G^2N^2$

In this section, the proposed generic framework is described and instantiated on the ML $(\mathcal{L}_3)$ fragment to generate our $G^2N^2$ model. As shown by Figure 1, 3 steps are involved:
**(1) defining the exhaustive CFG that generates the language, (2) reducing the exhaustive CFG, (3) translating the variables and the rules of the reduced CFG into GNN input and model layer.** To keep the expressive power of the language at each step, the equivalence between the successive representations must be ensured.

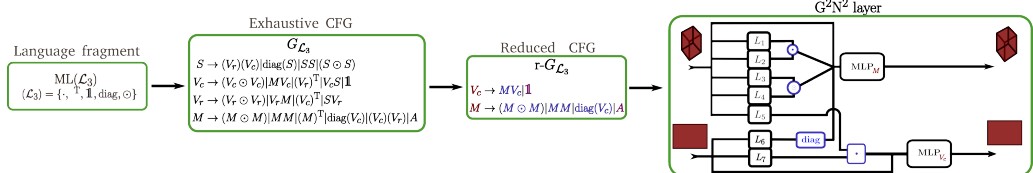

Figure 1: **Overview of the proposed framework instantiated on** ML $(\mathcal{L}_3)$.

### 3.1    FROM ML $(\mathcal{L}_3)$ TO THE EXHAUSTIVE CFG $G_{\mathcal{L}_3}$

The first step of the framework translates the language fragment into an exhaustive CFG (variables, terminal symbols and rules). For ML $(\mathcal{L}_3)$, the variables of the exhaustive CFG denoted $G_{\mathcal{L}_3}$ are defined using the following proposition proved in appendix A.2.

**Proposition 3.1**
*For any square matrix of size $n^2$, operations in $\mathcal{L}_3$ can only produce square matrices of the same size, row, or column vectors of size $n$ or scalars.*

In the context of our study, as in Geerts (2020), ML $(\mathcal{L}_3)$ is applied on the adjacency matrix. Thus, proposition 3.1 ensures that $G_{\mathcal{L}_3}$ variables are restricted to square matrix $(M)$, column vector $(V_c)$, row vector $(V_r)$ and scalar $(S)$. Once the variables defined, the production rules of $G_{\mathcal{L}_3}$ are obtained by enumerating all possible operations in ML $(\mathcal{L}_3)$ that produce such variables. The rule $M \to A$ is added in order to be compliant with Geerts (2020). All the rules composing $G_{\mathcal{L}_3}$ are synthesised in equation 1 where | denotes the classical OR operator since a variable can be produced by different rules. They fully characterise the CFG[3].

The following theorem ensures that the language generated by $G_{\mathcal{L}_3}$ is ML $(\mathcal{L}_3)$. Thus $G_{\mathcal{L}_3}$ is as expressive as ML $(\mathcal{L}_3)$.

**Theorem 3.1**
*For $G_{\mathcal{L}_3}$ defined by*

$$
\begin{aligned}
S &\to (V_r)(V_c) \mid \text{diag}(S) \mid SS \mid (S \odot S) \\
V_c &\to (V_c \odot V_c) \mid MV_c \mid (V_r)^{\mathbf{T}} \mid V_cS \mid \mathbb{1} \\
V_r &\to (V_r \odot V_r) \mid V_rM \mid (V_c)^{\mathbf{T}} \mid SV_r \\
M &\to (M \odot M) \mid MM \mid (M)^{\mathbf{T}} \mid \text{diag}(V_c) \mid (V_c)(V_r) \mid A
\end{aligned}
\tag{1}
$$

*we have*

$$
L(G_{\mathcal{L}_3}) = \text{ML}(\mathcal{L}_3).
$$

---

[3]Elements that are not variables in the rule set are said to be terminal symbols.

The full proof is provided in appendix (A.2). Its idea is the following. As any operation in the rules of $G_{\mathcal{L}_3}$ belongs to $\mathcal{L}_3$, it is clear that $L(G_{\mathcal{L}_3}) \subset \mathrm{ML}\,(\mathcal{L}_3)$. The reciprocal inclusion is proven by induction over the number of $\mathrm{ML}\,(\mathcal{L}_3)$ operations.

Given the results of theorem 3.1, the next step reduces the CFG by exploiting the redundancies in the exhaustive set of rules and variables.

## 3.2 From $G_{\mathcal{L}_3}$ to r-$G_{\mathcal{L}_3}$

An example of redundancy can be observed in the following proposition proved in the appendix (see A.2).

**Proposition 3.2**
*For any square matrix $M$, column vector $V_c$ and row vector $V_r$, we have*

$$M \odot (V_c \cdot V_r) = \mathrm{diag}\,(V_c)\, M \mathrm{diag}\,(V_r)$$

The following theorem guarantees that the following reduced grammar preserves expressiveness.

**Theorem 3.2** ($\mathrm{ML}\,(\mathcal{L}_3)$ reduced CFG )
*Let r-$G_{\mathcal{L}_3}$ be defined by*

$$
\begin{aligned}
V_c &\to MV_c \mid \mathbb{1} \\
M &\to (M \odot M) \mid MM \mid \mathrm{diag}\,(V_c) \mid A
\end{aligned}
\tag{2}
$$

*r-$G_{\mathcal{L}_3}$ is as expressive as $G_{\mathcal{L}_3}$.*

*Proof.* For any scalar $S, S'$, since $\mathrm{diag}\,(S)$, $S \odot S'$ and $S \cdot S'$ produce a scalar, the only way to produce a scalar from other variables is to pass through a vector dot product. Hence the scalar variable $S$ and its rules can be removed from $G_{\mathcal{L}_3}$ without loss of expressive power.

Since $\mathrm{diag}\,(v)\,w = v \odot w$ for any vector $v, w$, the vector Hadamard product can be removed from the vector rules. Proposition 3.2 allows to remove $V_c V_r$ from the rules of $M$ since the results of subsequent mandatory operations $MM$ or $MV_c$ can be obtained with other combinations. At this stage, the following intermediate CFG i-$G_{\mathcal{L}_3}$ is as expressive as $G_{\mathcal{L}_3}$ since it can compute any vector of $G_{\mathcal{L}_3}$.

$$
\begin{aligned}
V_c &\to MV_c \mid (V_r)^{\mathbf{T}} \mid \mathbb{1} \\
V_r &\to V_r M \mid (V_c)^{\mathbf{T}} \\
M &\to (M \odot M) \mid MM \mid (M)^{\mathbf{T}} \mid \mathrm{diag}\,(V_c) \mid A
\end{aligned}
$$

Since the remaining $M$ rules preserve symmetry, $(M)^{\mathbf{T}}$, the variable $V_r$ and its rules can be removed. It conducts to r-$G_{\mathcal{L}_3}$ defined in equation 2. $\qquad\square$

From these two steps, the resulting CFG r-$G_{\mathcal{L}_3}$ possesses the expressive power of the fragment $\mathrm{ML}\,(\mathcal{L}_3)$. The next step is a translation of r-$G_{\mathcal{L}_3}$ into a GNN layer.

## 3.3 From r-$G_{\mathcal{L}_3}$ to a $\mathrm{G}^2\mathrm{N}^2$ layer model

In r-$G(\mathcal{L}_3)$, any vector $V_c$ or matrix $M$ is produced by applying a sequence of rules on $A$ and $\mathbb{1}$. As a consequence, every matrix or vector can be attained through an iterative rule selection procedure using matrix and vector memories that store intermediate variables. Figure 2 describes this procedure: each iteration starts by choosing a rule in r-$G(\mathcal{L}_3)$ before selecting corresponding inputs in the memories. Applying the selected rule produces a new matrix or a new vector, which is added to the appropriate memory.

Translating this iterative procedure into a GNN based on a sequence of layers requires a memory management strategy and a selection mechanism for both rules and inputs, while taking into account learning issues related to downstream tasks.

The matrix memory aims at storing the variables $M$ produced by successive applications of r-$G(\mathcal{L}_3)$ rules. This memory is represented by a three order tensor $\mathcal{C}^{(l)}$ where produced matrices (i.e. edges embeddings in a GNN context) are stacked across layers on the third dimension. In the same way, the vector memory is dedicated to the variables $V_c$ that correspond to nodes embeddings. It is as a matrix $H^{(l)}$ where produced vectors are stacked on the second dimension. $\mathcal{C}^{(l)}$ and $H^{(l)}$ are the input of the $l$-th GNN layer which produces $\mathcal{C}^{(l+1)}$ and $H^{(l+1)}$ as output, as depicted in Figure 3 describing a G$^2$N$^2$ layer. While the memory of the iterative procedure grows with each iteration, a tractable GNN architecture constrains the stacking dimension to be set to a given value at each layer.

In order to mimic the rule selection procedure of Figure 2, a G$^2$N$^2$ layer applies a selection among the outputs produced by all the rules. Such a strategy enables to compute in parallel several occurrences of any rule with multiple inputs. Hence, parameterised quantities $b_{\odot}, b_{\cdot}, b_{\text{diag}}, b_{MV_c}$ of the rules $(M \odot M)$, $(MM)$, $\text{diag}\,(V_c)$, $MV_c$ are computed in parallel taking as input linear combination $L_i$ of slices of $\mathcal{C}^{(l)}$ and slices of $H^{(l)}$. These linear combinations are able to select among inputs $\mathcal{C}^{(l)}$ and $H^{(l)}$ through a learning paradigm.

Both the matrix rules outputs and the tensor $\mathcal{C}^{(l)}$ (obtained through a skip connection which guarantees the memory persistence) are fed to $\text{MLP}_M$ that produces the output tensor $\mathcal{C}^{(l+1)}$ with a selected third dimension size $S^{(l+1)}$. This MLP allows in the same time to simulate the rule selection, to compress the matrix output of the layer to a fixed size and to learn a point wise function for solving specific downstream tasks. It relates to the set of operations $\{+, \times, f\}$ of $MATLANG$ and does not modify the expressive power (Geerts (2020); Maron et al. (2019a)). The output $H^{(l+1)}$ is provided similarly through $\text{MLP}_{V_c}$. Figure 3 describes the whole model of a G$^2$N$^2$ layer.

Formally, the update equations are :

$$\mathcal{C}^{(l+1)} = \text{MLP}_M(\mathcal{C}^{(l)}||L_1(\mathcal{C}^{(l)}) \cdot L_2(\mathcal{C}^{(l)})||L_3(\mathcal{C}^{(l)}) \odot L_4(\mathcal{C}^{(l)})||\text{diag}(L_6(H^{(l)}))), \qquad (3)$$

$$H^{(l+1)} = \text{MLP}_{V_c}(H^{(l)}||L_5(\mathcal{C}^{(l)}) \cdot L_7(H^{(l)})), \qquad (4)$$

where $||$ is the concatenation. $\text{MLP}_M$ and $\text{MLP}_{V_c}$ are learnable MLPs, and $L_i$ are learnable linear blocks acting on the third dimension of $\mathcal{C}^{(l)}$ or the second dimension of $H^{(l)}$: $L_{1,2}$ : $\mathbb{R}^{S^{(l)}} \to \mathbb{R}^{b_{\cdot}^{(l)}}$, $L_{3,4} : \mathbb{R}^{S^{(l)}} \to \mathbb{R}^{b_{\odot}^{(l)}}$, $L_5 : \mathbb{R}^{S^{(l)}} \to \mathbb{R}^{b_{MV_c}^{(l)}}$, $L_6 : \mathbb{R}^{f^{(l)}} \to \mathbb{R}^{b_{\text{diag}}^{(l)}}$, $L_7 : \mathbb{R}^{f^{(l)}} \to \mathbb{R}^{b_{MV_c}^{(l)}}$, $\text{MLP}_M : \mathbb{R}^{S^{(l)}+b_{\cdot}^{(l)}+b_{\odot}^{(l)}+b_{\text{diag}}^{(l)}} \to \mathbb{R}^{S^{(l+1)}}$, and $\text{MLP}_{V_c} : \mathbb{R}^{f^{(l)}+b_{MV_c}^{(l)}} \to \mathbb{R}^{f^{(l+1)}}$.

### 3.4 G$^2$N$^2$ architecture and its expressive power

Figure 4 depicts the global G$^2$N$^2$ architecture. The inputs are $H^{(0)}$ and $\mathcal{C}^{(0)}$. $H^{(0)}$ of size $n \times f_n + 1$ is the feature nodes matrix concatenated with $\mathbb{1}$. $\mathcal{C}^{(0)} \in \mathbb{R}^{n \times n \times (f_e+1)}$ is a stacking on the third dimension of the adjacency matrix $A$ and the extended adjacency tensor $E$ of size $n \times n \times f_e$, where $f_e$ is the number of edge features.

After the last layer, permutation equivariant readout functions are applied on both $H^{(l_{\text{end}})}$ and the diagonal and off-diagonal components of $\mathcal{C}^{(l_{\text{end}})}$. Readout outputs are then fed to a dedicated decision layer.

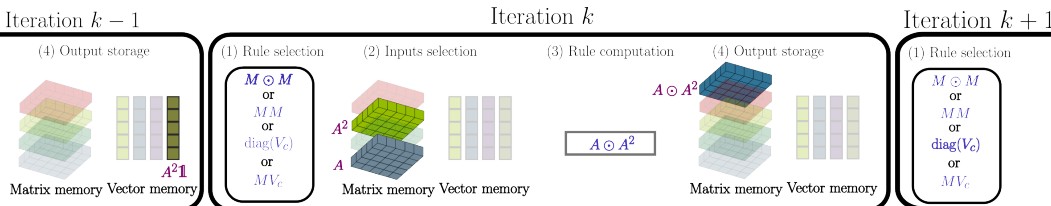

Figure 2: **4-step iterative procedure** (1) Rule selection (2) Inputs selection: inputs relative to the chosen rule are selected from matrix and/or vector memories (opaque matrices) (3) Rule computation (4) Output storage: the produced output is stored into its relative memory.

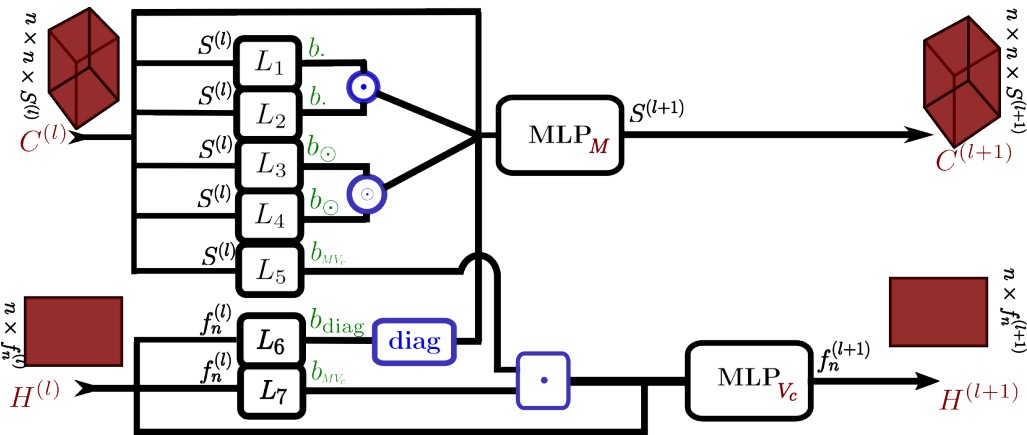

Figure 3: $L_1$-$L_5$ combine the $S^{(l)}$ slices of $\mathcal{C}^{(l)}$ into $2b_\odot$, $2b_\cdot$ and $b_{MV_c}$ matrices. $L_6$-$L_7$ combine the $f_n^{(l)}$ columns of $H^{(l)}$ into $b_{\mathrm{diag}}$ and $b_{MV_c}$ vectors. From the outputs of $L_1$-$L_7$, multiple occurrences of r-$G(\mathcal{L}_3)$ rules $(M \odot M)$, $(MM)$, $(\mathrm{diag}(V_c)$ and $(MV_c)$ are computed. The obtained outputs and the layer inputs are fed to $\mathrm{MLP}_M$ and $\mathrm{MLP}_{V_c}$ providing the layer outputs $\mathcal{C}^{(l+1)}$ and $H^{(l+1)}$.

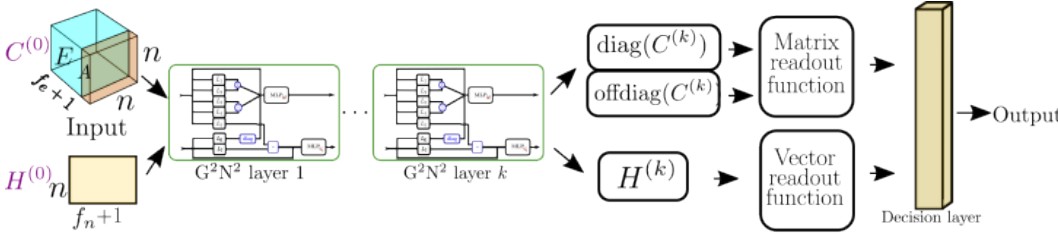

Figure 4: **Model of $G^2N^2$ architecture from the graph to the output**. Each layer updates node and edge embeddings and readout functions are applied independently on $H^{(k)}$ and the diagonal and the non-diagonal elements of $\mathcal{C}^{(k)}$.

**Theorem 3.3** (Expressive power of $G^2N^2$)
*$G^2N^2$ is able to produce any matrix and vector of $L(r$-$G_{\mathcal{L}_3})$. It is as expressive as 3-WL.*

*Proof.* We show that $G^2N^2$ at layer $l$ can produce all matrices and vectors r-$G_{\mathcal{L}_3}$ can produce, after $l$ iterations. It is true for $l = 1$. Indeed, at r-$G_{\mathcal{L}_3}$ first iteration, we obtain the matrices I, $A$, $A^2$ and the vectors $\mathbb{1}$ and $A\mathbb{1}$. Since any of $L_i(\mathcal{C}^{(0)})$ for $i \in [\![1, 5]\!]$ is a linear combination of $A$ and I, $G^2N^2$ can produce those vectors and matrices in one layer.

Suppose that there exists $l > 0$ such that $G^2N^2$ can produce any of the matrices and vectors r-$G_{\mathcal{L}_3}$ can after $l$ iterations. We denote by $\mathcal{A}_l$ the set of those matrices and by $\mathcal{V}_l$ the set of those vectors. At the $l+1$-th iteration, we have $\mathcal{A}_{l+1} = \{M \odot N, MN, \mathrm{diag}(V_c) | M, N \in \mathcal{A}_l \, V_c \in \mathcal{V}_l\}$ and $V_{l+1} = \{MV_c | M \in \mathcal{A}_k, V_c \in \mathcal{V}_l\}$. Let $M, N \in \mathcal{A}_l$ and $V_c \in \mathcal{V}_l$ then by hypothesis $G^2N^2$ can produce $M, N$ at layer $l$. Since $L$ produces at least two different linear combinations of matrices or vectors in respectively $\mathcal{A}_l$ and $\mathcal{V}_l$, $MN$, $M \odot N$, $MV_c$ and $\mathrm{diag}(V_c)$ are reachable at layer $l + 1$. Thus $\mathcal{A}_{l+1}$ is included in the set of matrices $G^2N^2$ can produce at layer $l + 1$ and $V_{l+1}$ is included in the set of vectors $G^2N^2$ can produce at layer $l + 1$. $\square$

## 3.5 DISCUSSION : $G^2N^2$ IN THE 3-WL GNN LITERATURE

**Positioning w.r.t Maron et al. (2019a)** From PPGN layer description (see Figure 2 of Maron et al. (2019a)), one can build the following CFG:

$$M \to MM \mid \mathrm{diag}(\mathbb{1}) \mid A \tag{5}$$

where $M \to \mathrm{diag}\,(\mathbb{1})$ and $M \to A$ represent inputs of the architecture as for $\mathrm{G}^2\mathrm{N}^2$. Compared to r-$G_{\mathcal{L}_3}$, $V_c$ variable and $M \to M \odot M$, $\mathrm{diag}\,(V_c)$ and $MV_c$ rules are missing. As a consequence, PPGN 3-WL expressive power is not formally inherited from $\mathrm{ML}\,(\mathcal{L}_3)$. As stated in the introduction, it relies on PPGN ability to mimic 2-FWL colouring and hashing steps. Its capacity to implement the colouring step relies on MLP universality. It explains that PPGN can approximate the missing rules of r-$G_{\mathcal{L}_3}$. To guarantee such an approximation, a certain width and depth for MLP are needed. $\mathrm{G}^2\mathrm{N}^2$ does not suffer from these computational constraints since it only needs to provide linear combinations as arguments of the operations.

3-IGN processes on sets of third order tensors. As a consequence, it cannot be described by a CFG derived from $\mathrm{ML}\,(\mathcal{L}_3)$. However, we can connect our approach with $k$-IGN. For $k$-IGN, the expressive power is related to MLPs and to the basis of linear equivariant operators defined in Maron et al. (2019b). In some ways, these operators can be linked to the algebraic operations of our framework. An example of such a link is given in appendix A.3 for 2-IGN .

**Positioning w.r.t Balcilar et al. (2021)** In appendix A.1, we show that GNNML1 (Balcilar et al. (2021)) can be seen as the resulting GNN of our framework applied on $\mathrm{ML}\,(\mathcal{L}_1)$. Concerning GNNML3, a CFG can also be deduced from its layer

$$V_c \to C_1 V_c \mid \ \cdots \ \mid C_k V_k \mid V_c \odot V_c \mid \mathbb{1}$$

where the matrices $C_1, \cdots, C_k$ are defined using the adjacency matrix, exponential pointwise function, and matrix Hadamard product. As some rules and variables are missing compared to r-$G_{\mathcal{L}_3}$, it cannot formally inherit the expressive power of $\mathrm{ML}\,(\mathcal{L}_3)$.

## 4 Experiments

This section is dedicated to the experimental validation of both the framework and $\mathrm{G}^2\mathrm{N}^2$. It answers 4 questions **Q1**-**Q4**. **Q1** concerns the validation of the reduced grammars. **Q2** and **Q3** relate to performance of $\mathrm{G}^2\mathrm{N}^2$ on downstream regression/classification tasks. **Q4** concerns the model spectral ability. Experimental settings are detailed in appendix C.

**Q1**: Is the reduction of grammar relevant and optimal?

This experiment aims at investigating the impact of the CFG reduction scheme through the comparison of different models built using $G_{\mathcal{L}_3}$ (see Figure 9 in appendix A), i-$G_{\mathcal{L}_3}$ (see Figure 10 in appendix A) and r-$G_{\mathcal{L}_3}$ (see Figure 3). The comparison is completed by an ablation study the aim of which is to investigate the importance of each rule of r-$G_{\mathcal{L}_3}$.

We use a graph regression benchmark called QM9 which is composed of 130K small molecules (Ramakrishnan et al. (2014); Wu et al. (2018)). For this study, we focus on the regression target $R^2$, which is known to be the most difficult to predict. As in Maron et al. (2019a), the dataset is randomly split into training, validation, and test sets with a respective ratio of 0.8, 0.1 and 0.1. The edge and vector embeddings are always of size 32.

The results are presented in Figure 5 where each model is represented in a 2-D space using the Mean Absolute Error (MAE) of the best validation model on the test set and the number of parameters of the model. These results corroborate the theoretical results of section 3, discussed in greater detail in appendix A.5: the MAE scores are comparable for $G(\mathcal{L}_3)$, i-$G(\mathcal{L}_3)$ and r-$G(\mathcal{L}_3)$ while the number of parameters is divided by 2 when reducing from $G(\mathcal{L}_3)$ to r-$G(\mathcal{L}_3)$. As expected, removing rules in r-$G(\mathcal{L}_3)$ leads to a drop of MAE performance. It also offers insights into the weights of each operation in the model and enables informed pruning of the GNN if the expressiveness is not required.

**Q2**: Does $\mathrm{G}^2\mathrm{N}^2$ perform better than other 3-WL GNNs for regression?

For this second question, we also use the dataset QM9, but we consider the 12 regression targets. The dataset is randomly split into training, validation, and test sets with the same ratio as in **Q1**. The experimental settings are detailed in appendix C. $\mathrm{G}^2\mathrm{N}^2$ results are compared to those in Huang et al. (2023); Maron et al. (2019a) including 1-GNN and

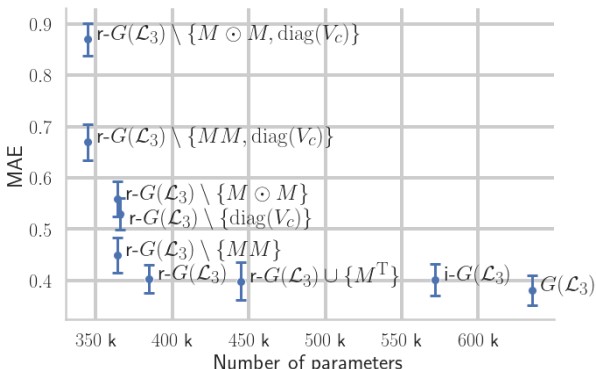

Figure 5: Comparison of model size and MAE performance on the QM9 $R^2$ target for GNNs derived from $G(\mathcal{L}_3)$. Each GNN model is denoted by its set of rules. Over-reduced grammar from r-$G(\mathcal{L}_3)$ are denoted with a $\setminus$, whereas $\cup$ denotes the addition of a rule to the set.

1-2-3-GNN (Morris et al. (2019)), DTNN (Wu et al. (2018)), DeepLRP (Chen et al. (2020)), NGNN (Zhang and Li (2021)), $I^2$-GNN (Huang et al. (2023)) and PPGN Maron et al. (2019a). The metric is the MAE of the best validation model on the test set. The mean epoch duration is measured on the same device for comparison between $G^2N^2$ and PPGN.

As in Maron et al. (2019a), we made two experiments. The first one consists in learning one target at a time while the second learns every target at once. In the first experiment, we have $S^{(l)} = f_n^{(l)} = 64$ and in the second $S^{(l)} = f_n^{(l)} = 32$. Partial results focusing on the two best models are given in Table 1. Complete results and experiment settings are given in appendix C. In both cases, $G^2N^2$ obtains the best results while learning faster.

Table 1: Results on QM9 dataset focusing on the best methods. On the left part, each target is learned separately while on the right side all targets are learned at the same time. The metric is MAE, the lower, the better. Complete results can be found in Table 4.

| Target | PPGN | $G^2N^2$ | PPGN | $G^2N^2$ |
|---|---|---|---|---|
| $\mu$ | 0.0934 | **0.0703** | 0.231 | **0.102** |
| $\alpha$ | 0.318 | **0.127** | 0.382 | **0.196** |
| $\epsilon_{homo}$ | 0.00174 | **0.00172** | 0.00276 | **0.0021** |
| $\epsilon_{lumo}$ | 0.0021 | **0.00153** | 0.00287 | **0.00211** |
| $\Delta\epsilon$ | 0.0029 | **0.00253** | 0.0029 | **0.00287** |
| $R^2$ | 3.78 | **0.342** | 16.07 | **1.19** |
| ZPVE | 0.000399 | **0.0000951** | 0.00064 | **0.0000151** |
| $U_0$ | 0.022 | **0.0169** | 0.234 | **0.0502** |
| $U$ | 0.0504 | **0.0162** | 0.234 | **0.0503** |
| $H$ | 0.0294 | **0.0176** | 0.229 | **0.0503** |
| $G$ | 0.024 | **0.0214** | 0.238 | **0.0504** |
| $C_v$ | 0.144 | **0.0429** | 0.184 | **0.0707** |
| T / ep | 129 s | 98 s | 131 s | 57 s |

**Q3**: Does $G^2N^2$ perform better than other 3-WL GNNs for classification?

For graph classification, we evaluate $G^2N^2$ on the classical TUD benchmark (Morris et al. (2020)), using the evaluation protocol of Xu et al. (2019). Results of GNNs and Graph Kernel are taken from Bouritsas et al. (2022). Since the number of node and edge features is very different from one dataset to another, the parameter settings for each of the 6 experiments related to these datasets can be found in Table 5 of appendix C. Partial results focusing on $G^2N^2$ performance are given in Table 2. Complete results can be seen in Table 6 of appendix C. $G^2N^2$ achieves better than rank 2 for five of the six datasets.

Table 2: Results of $G^2N^2$ on TUD dataset compared to the best GNN competitor. The rank of $G^2N^2$ within GNNs is in parentheses. The metric is accuracy, the higher, the better. Complete results can be seen in Table 6.

| Dataset | $G^2N^2$ | rank | Best GNN competitor |
|---------|----------|------|---------------------|
| MUTAG | 92.5±5.5 | 1(1) | 92.2±7.5 |
| PTC | 72.3±6.3 | 1(1) | 68.2±7.2 |
| Proteins | 80.1±3.7 | 1(1) | 77.4±4.9 |
| NCI1 | 82.8±0.9 | 5(3) | 83.5±2.0 |
| IMDB-B | 76.8±2.8 | 3(3) | 77.8±3.3 |
| IMDB-M | 54.0±2.9 | 2(2) | 54.3±3.3 |

**Q4**: CAN $G^2N^2$ LEARN BAND-PASS FILTERS IN THE SPECTRAL DOMAIN?

As shown in Balcilar et al. (2020), the spectral ability of a GNN and particularly its ability to operate as band-pass filter is an important property of a model for certain downstream tasks. In order to assess the spectral ability of $G^2N^2$ and answer Q4, we use the protocol and node regression dataset of Balcilar et al. (2021). $R^2$ score is used to compare performance.

Table 3 reports the comparison of $G^2N^2$ to CHEBNET (Hammond et al. (2011)), PPGN and GNNML3, citing the results from Balcilar et al. (2021). CHEBNET and GNNML3 are spectrally designed and manage to learn low-pass, high-pass, and band-pass filters. For the three filter types, $G^2N^2$ reaches comparable performance. In appendix B, a theoretical analysis shows that a 3-WL GNN is able to approximate any type of filter.

As shown in the table, PPGN fails to learn band-pass filters. This result which contradicts the previous theoretical result is related to memory and complexity issues. Hence, as explained before, PPGN needs a deeper and wider architecture for this task that can not be reached for 900 node graphs (Balcilar et al. (2021)).

Table 3: $R^2$ score on spectral filtering node regression. Results are a median of 10 runs.

| Method | Low-pass | High-pass | Band-pass |
|--------|----------|-----------|-----------|
| CHEBNET | 0.9995 | 0.9901 | 0.8217 |
| GNNML3 | 0.9995 | 0.9909 | 0.8189 |
| PPGN | 0.9991 | **0.9925** | 0.1041 |
| $G^2N^2$ | **0.9996** | **0.9994** | 0.8206 |

## 5 CONCLUSION

Designing provably expressive GNNs has been the target of many recent works. In this paper, we have proposed a new theoretical framework for designing such models. Taking as input a language fragment, i.e. a set of algebraic operations, the framework uses reduced Context Free Grammars to drive the generation of graph neural architectures with provable expressive power. The framework provides insights about the importance of algebraic operations in the resulting model, as shown by the experimental grammar ablative study. Such results can be useful for improving the performance vs. computational cost trade-off for a given task.

Through the application of the framework to ML $(\mathcal{L}_3)$ fragment, the paper also proposed the provably 3-WL $G^2N^2$ model. In addition to these theoretical guarantees, $G^2N^2$ is also shown to be efficient for solving graph learning downstream tasks through several experiments on regression, classification and spectral filtering benchmarks. In all cases, $G^2N^2$ outperforms 3-WL GNNs, while being more tractable.

Beyond these results, we are convinced that our contributions open the door to the design of models surpassing 3-WL, taking as root a language manipulating tensors of greater order (Geerts and Reutter (2022)). Moreover, the framework is not limited to GNN models since many other learning paradigm can be modeled with algebraic languages.

## Acknoledgements

The authors acknowledge the support of the French Agence Nationale de la Recherche (ANR) under grant ANR-21-CE23-0025 (CoDeGNN project). The authors acknowledge the support of the ANR and the Région Normandie under grant ANR-20-THIA-0021 (HAISCoDe project).

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

This appendices provide additional content to the main paper $G^2N^2$: Weisfeiler and Lehman go grammatical.

## A CFG and GNN

### A.1 From ML $(\mathcal{L}_1)$ to 1-WL GNN

In this subsection, the reduction framework described in section 3 is applied to the fragment ML $(\mathcal{L}_1)$ as shown by Figure 6.

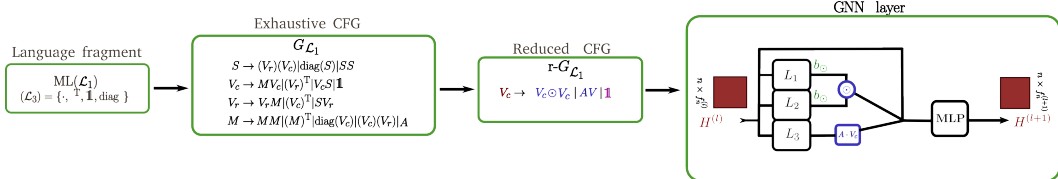

Figure 6: Overview of the proposed framework instantiated on ML $(\mathcal{L}_1)$.

To determine the variables of the CFG, the following proposition is necessary.

**Proposition A.1**
*For any square matrix of size $n^2$, operations in $\mathcal{L}_1$ can only produce square matrices of size $n^2$, row or column vectors of size $n$ or scalars.*

*Proof.* Let $M$ be a square matrix of size $n^2$, we first need to prove that $\mathcal{L}_1$ can produce square matrices of size $n^2$, row and column vectors of size $n$ and scalars.

Yet $\mathbb{1} := \mathbb{1}(M)$ is a column vector of size $n$, $\mathbb{1}^{\mathbf{T}}$ is a row vector of size $n$, $\mathbb{1}^{\mathbf{T}} \cdot \mathbb{1}$ is a scalar and $M$ is a square matrix of size $n^2$.

Then let $N \in \mathbb{R}^{n \times n}$, $v \in \mathbb{R}^n$, $w \in (\mathbb{R}^n)^*$, and $s \in \mathbb{R}$ be words of ML $(\mathcal{L}_1)$, we have

$$
\begin{array}{llll}
M \cdot N \in \mathbb{R}^{n \times n} & M \cdot v \in \mathbb{R}^n & w \cdot M \in (\mathbb{R}^n)^* & w \cdot v \in \mathbb{R} \\
v \cdot w \in \mathbb{R}^{n \times n} & \mathbb{1}(v) \in \mathbb{R}^n & v^{\mathbf{T}} \in (\mathbb{R}^n)^* & \mathbb{1}(w) \in \mathbb{R} \\
M^{\mathbf{T}} \in \mathbb{R}^{n \times n} & w^{\mathbf{T}} \in \mathbb{R}^n & s \cdot w \in (\mathbb{R}^n)^* & \mathrm{diag}(s) \in \mathbb{R} \\
\mathrm{diag}(v) \in \mathbb{R}^{n \times n} & \mathbb{1}(M) \in \mathbb{R}^n & & s \cdot s \in \mathbb{R} \\
& v \cdot s \in \mathbb{R}^n & & \mathbb{1}(s) \in \mathbb{R}
\end{array}
$$

Since this is an exhaustive list of all operations ML $(\mathcal{L}_1)$ can produce with these words, we can conclude. $\square$

**Theorem A.1** (ML $(\mathcal{L}_1)$ reduced CFG )
*The following CFG denoted $r$-$G_{\mathcal{L}_1}$ is as expressive as 1-WL.*

$$V_c \to \mathrm{diag}(V_c) V_c \mid AV_c \mid \mathbb{1} \tag{6}$$

*Proof.* Proposition A.1 leads to only four variables. $M$ for the square matrices, $V_c$ for the column vectors, $V_r$ for the row vectors and $S$ for the scalars. We define a CFG $G_{\mathcal{L}_1}$ where the rules of a given variable is every possible operation in ML $(\mathcal{L}_1)$ that produce this variable:

$$
\begin{aligned}
S &\to (V_r)(V_c) \mid \mathrm{diag}(S) \mid SS \\
V_c &\to MV_c \mid (V_r)^{\mathbf{T}} \mid V_c S \mid \mathbb{1} \\
V_r &\to V_r M \mid (V_c)^{\mathbf{T}} \mid SV_r \\
M &\to MM \mid (M)^{\mathbf{T}} \mid \mathrm{diag}(V_c) \mid (V_c)(V_r) \mid A
\end{aligned}
\tag{7}
$$

As any operation in the rules of $G_{\mathcal{L}_1}$ belongs to $\mathcal{L}_1$, it is clear that $L(G_{\mathcal{L}_1}) \subset \text{ML}(\mathcal{L}_1)$. For the other inclusion, a simple inductive proof on the maximal number of rules shows that any sentence produced by $\text{ML}(\mathcal{L}_1)$ can be derived from $G_{\mathcal{L}_1}$. We have then $\text{ML}(\mathcal{L}_1) = L(G_{\mathcal{L}_1})$. For any scalar $s, s'$, since $\text{diag}(s)$ and $s \cdot s'$ produce a scalar, the only way to produce a scalar from another variable is to pass through a vector dot product. It implies that to generate scalars, we only need to be able to generate vectors. We can then reduce $G_{\mathcal{L}_1}$ by removing the scalar variable $S$ and setting $V_c$ as starting variable.

$$V_c \to MV_c \mid (V_r)^{\mathbf{T}} \mid \mathbb{1}$$
$$V_r \to V_r M \mid (V_c)^{\mathbf{T}}$$
$$M \to MM \mid (M)^{\mathbf{T}} \mid \text{diag}(V_c) \mid (V_c)(V_r) \mid A$$

To ensure that the start variable is $V_c$, a mandatory subsequent operation will be $MV_c$ for any matrix variable $M$. As a consequence, by associativity of the matrix multiplication, $MM$ and $(V_c)(V_r)$ can be removed from the rule of $M$.

$$V_c \to MV_c \mid (V_r)^{\mathbf{T}} \mid \mathbb{1}$$
$$V_r \to V_r M \mid (V_c)^{\mathbf{T}}$$
$$M \to (M)^{\mathbf{T}} \mid \text{diag}(V_c) \mid A$$

Since diag produces symmetric matrices and $A$ is symmetric, $(M)^{\mathbf{T}}$ does not play any role here. As a consequence, we can then focus on the column vector and we obtain r-$G_{\mathcal{L}_1}$. $\quad\square$

Figure 7 shows how the CFG $G_{\mathcal{L}_1}$ produces the sentence $\mathbb{1}^{\mathbf{T}} A \mathbb{1}$.

Figure 7: $G_{\mathcal{L}_1}$ generating the sentence $\mathbb{1}^{\mathbf{T}} A \mathbb{1}$. From the starting variable, Variables are replaced by applying rules until only terminal symbols remain.

## A.2    Proofs of section 3

This subsection is dedicated to proof of propositions and theorem of section 3.

Firstly, we prove proposition 3.1.

*Proof.* Since $\mathcal{L}_1 \subset \mathcal{L}_3$, we only need to check the rule associated with the matrix Hadamard product can produce. Let $M \in \mathbb{R}^{n \times n}$ and $N \in \mathbb{R}^{n \times n}$ be words $\text{ML}(\mathcal{L}_3)$ can produce, we have $M \odot N \in \mathbb{R}^{n \times n}$. We can conclude. $\quad\square$

Secondly, we prove proposition 3.2.

*Proof.* Let $M$ be a square matrix, $V_c, V_r$ be respectively column and row vectors, we have for any $i, j$,

$$
\begin{aligned}
(M \odot (V_c \cdot V_r))_{i,j} &= M_{i,j}(V_c \cdot V_r)_{i,j} \\
&= (V_c)_i M_{i,j}(V_r)_j \\
&= \sum_l \text{diag}(V_c)_{i,l} M_{l,j}(V_r)_j \\
&= (\text{diag}(V_c) M)_{i,j}(V_r)_j \\
&= \sum_l (\text{diag}(V_c) M)_{i,l}\text{diag}(V_r)_{l,j} \\
&= (\text{diag}(V_c) M \text{diag}(V_r))_{i,j}
\end{aligned}
$$

We only use the scalar product commutativity here. $\quad\square$

Eventually, we prove theorem 3.1.

*Proof.* As any operation in the rules of $G_{\mathcal{L}_3}$ belongs to $\mathcal{L}_3$, it is clear that $L(G_{\mathcal{L}_3}) \subset \mathrm{ML}\,(\mathcal{L}_3)$.

Let $k$ be a positive integer, we denote by $M_{\mathcal{L}_3}^k$, $Vc_{\mathcal{L}_3}^k$, $Vr_{\mathcal{L}_3}^k$ and $S_{\mathcal{L}_3}^k$ the set of matrices, column vectors, row vectors and scalars that can be produce with at most $k$ operation in $\mathcal{L}_3$ from $A$. We also denote by $M_G^k$, $Vc_G^k$, $Vr_G^k$ and $S_G^k$ the set of matrices, column vectors, row vectors and scalars that can be produce with at most $k$ rules applied in $G_{\mathcal{L}_3}$.

For $k = 0$, we have $Vc_{\mathcal{L}_3}^0 = Vr_{\mathcal{L}_3}^0 = S_{\mathcal{L}_3}^0 = \emptyset$, and thus $Vc_{\mathcal{L}_3}^0 \subset Vc_G^0$, $Vr_{\mathcal{L}_3}^0 \subset Vr_G^0$ and $S_{\mathcal{L}_3}^0 \subset S_G^0$. Moreover $M_{\mathcal{L}_3}^0 = \{A\}$ and $M_G^0 = \{A\}$.

Let suppose that there exists $k \geqslant 0$ such that $M_{\mathcal{L}_3}^k \subset M_G^k$, $Vc_{\mathcal{L}_3}^k \subset Vc_G^k$, $Vr_{\mathcal{L}_3}^k \subset Vr_G^k$ and $S_{\mathcal{L}_3}^k \subset S_G^k$. Then since $G_{\mathcal{L}_3}$ rules is composed of the exhaustive operations in $\mathcal{L}_3$, we have that $M_{\mathcal{L}_3}^{k+1} \subset M_G^{k+1}$, $Vc_{\mathcal{L}_3}^{k+1} \subset Vc_G^{k+1}$, $Vr_{\mathcal{L}_3}^{k+1} \subset Vr_G^{k+1}$ and $S_{\mathcal{L}_3}^{k+1} \subset S_G^{k+1}$ By induction, we have that $L(G_{\mathcal{L}_3}) \subset \mathrm{ML}\,(\mathcal{L}_3)$ and we can conclude that $L(G_{\mathcal{L}_3}) = \mathrm{ML}\,(\mathcal{L}_3)$. $\qquad\square$

### A.3 CFG to describe existent architectures

The following examples show how CFG can be used to characterise GNNs.

**Proposition A.2** (GCN CFG)
*The following CFG, strictly less expressive than* $\mathrm{ML}\,(\mathcal{L}_1)$*, represents GCN (Kipf and Welling (2017))*

$$V_c \to CV_c \mid \mathbb{1} \tag{8}$$

*where* $C = \mathrm{diag}\,((A+I)\mathbb{1})^{-\frac{1}{2}}\,(A+I)\mathrm{diag}\,((A+I)\mathbb{1})^{-\frac{1}{2}}$

In GCN, the only grammatical operation is the message passing given by $CV_c$ where $C$ is the convolution support. The other parts of the model are linear combinations of vectors and MLP, that correspond to $+,\times,$ and $f$ in the language. Since $+,\times,$ and $f$ do not affect the expressive power of the language (Geerts (2020)), they do not appear in the grammar. Actually, any MPNN based on $k$ convolution support $C_i$ included in $\mathrm{ML}\,(\mathcal{L}_1)$ can be described by the following CFG which is strictly less expressive than $\mathrm{ML}\,(\mathcal{L}_1)$:

$$V_c \to C_1 V_c \mid \cdots \mid C_k V_c \mid \mathbb{1} \tag{9}$$

GNNML1 is an architecture provably 1-WL equivalent (Balcilar et al. (2021)) with the following node update.

$$\begin{aligned}
H^{(l+1)} = {} & H^{(l)}W^{(l,1)} + AH^{(l)}W^{(l,2)} \\
& + H^{(l)}W^{(l,3)} \odot H^{(l)}W^{(l,1)}.
\end{aligned} \tag{10}$$

Where $H^{(l)}$ is the matrix of node embedding at layer $l$ and $W^{(l,i)}$ are learnable weight matrices. For any vector $v, w$, since $\mathrm{diag}\,(v)\,w = v \odot w$, the following CFG that describes GNNML1 is equivalent to r-$G_{\mathcal{L}_1}$.

$$V_c \to V_c \odot V_c \mid AV_c \mid \mathbb{1} \tag{11}$$

**From r-$G_{\mathcal{L}_3}$ to MPNNs and PPGN** We have already shown that most MPNNs can be written with operations in r-$G_{\mathcal{L}_1}$, since $\mathcal{L}_1 \subset \mathcal{L}_3$ it stands for r-$G_{\mathcal{L}_3}$. PPGN can also be written with r-$G_{\mathcal{L}_3}$. Indeed, at each layer, PPGN applies the matrix multiplication on matched matrices on the third dimension, an operation included in r-$G_{\mathcal{L}_3}$. The node features are stacked on the third dimension as diagonal matrices, the diag operation is also included in r-$G_{\mathcal{L}_3}$. As all operations in PPGN are included, r-$G_{\mathcal{L}_3}$ generalises PPGN layer. Actually, the following CFG describes the PPGN layer :

$$M \to MM \mid \mathrm{diag}\,(\mathbb{1}) \mid A \tag{12}$$

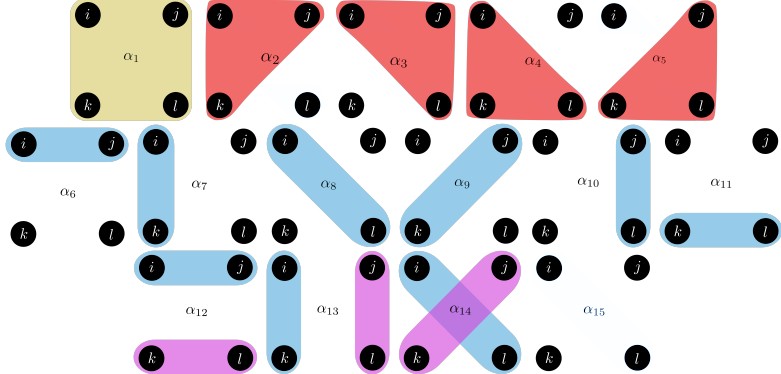

Figure 8: Partition of four indices tuples.

**2-IGN CFG** For $p \in [\![1,15]\!]$, we define $\mathcal{B}_p \in (\mathbb{R}^n)^4$ as follow, $\mathcal{B}_p = \begin{cases} 1 \text{ if } (i,j,k,l) \in \alpha_p, \\ 0 \text{ if not.} \end{cases}$.

Where $(\alpha_p)$ corresponds to the 15 manners to partition four elements that can be seen in Figure 8.

As shown in Maron et al. (2019b), $(\mathcal{B}_p)$ is a basis of the set of equivariant linear operators from $(\mathbb{R}^n)^2$ to $(\mathbb{R}^n)^2$. For the proof in the paper, two isomorphisms $vec : (\mathbb{R}^n)^2 \to \mathbb{R}^{n^2}$ and $mat : (\mathbb{R}^n)^4 \to (\mathbb{R}^{n^2})^2$ was defined for any tensor $T \in (\mathbb{R}^n)^4$, matrices $M \in (\mathbb{R}^{n^2})^2$, $N \in (\mathbb{R}^n)^2$ and vector $v \in \mathbb{R}^{n^2}$.

$$mat(T)_{i,j} = T_{i//n, i\%n, j//n, j\%n}$$
$$mat^{-1}(M)_{i,j,k,l} = M_{in+j, kn+l}$$
$$vec(N)_i = N_{i//n, i\%n}$$
$$vec^{-1}(v)_{i,j} = v_{in+j}$$

We can then define the binary operation $\tilde{\cdot}$ as follow

$$T \tilde{\cdot} N = vec^{-1}(mat(T)vec(N))$$

Actually, we obtain the following operation

$$(T \tilde{\cdot} N)_{i,j} = \sum_{k,l} T_{i,j,k,l} N_{k,l}$$

We have all we need to proceed on writing 2-IGN as a grammar. The idea is to compute the basis operator to any matrices with a set of rules.

$$(\mathcal{B}_1 \tilde{\cdot} N)_{i,j} = \sum_{k,l} (\mathcal{B}_1)_{i,j,k,l} N_{k,l}$$

$$= \begin{cases} N_{i,i} \text{ if } i = j, \\ 0 \text{ if not.} \end{cases}$$

It is pretty easy to see that

$$\mathcal{B}_1 \tilde{\cdot} N = N \odot \mathrm{I}$$

$$(\mathcal{B}_2 \tilde{\cdot} N)_{i,j} = \sum_{k,l} (\mathcal{B}_2)_{i,j,k,l} N_{k,l}$$

$$= \begin{cases} \sum_{l \neq i} N_{i,l} \text{ if } i = j, \\ 0 \text{ if not.} \end{cases}$$

Here, it is a sum over the matrix line avoiding the diagonal.

$$\mathcal{B}_2\tilde{\phantom{.}}N = \mathrm{diag}\left((N \odot J)\mathbb{1}\right)$$

$$(\mathcal{B}_3\tilde{\phantom{.}}N)_{i,j} = \sum_{k,l}(\mathcal{B}_3)_{i,j,k,l}N_{k,l}$$

$$= \begin{cases} \sum_{l \neq i} N_{l,i} \text{ if } i = j, \\ 0 \text{ if not.} \end{cases}$$

Here, it is a sum over the matrix column avoiding the diagonal.

$$\mathcal{B}_3\tilde{\phantom{.}}N = \mathrm{diag}\left((N \odot J)^{\mathbf{T}}\mathbb{1}\right)$$

$$(\mathcal{B}_4\tilde{\phantom{.}}N)_{i,j} = \sum_{k,l}(\mathcal{B}_4)_{i,j,k,l}N_{k,l}$$

$$= \begin{cases} N_{j,j} \text{ if } i \neq j, \\ 0 \text{ if not.} \end{cases}$$

It is the projection of the corresponding column diagonal element.

$$\mathcal{B}_4\tilde{\phantom{.}}N = (\mathbb{1}\mathbb{1}^{\mathbf{T}}(N \odot \mathrm{I})) \odot J$$

$$(\mathcal{B}_5\tilde{\phantom{.}}N)_{i,j} = \sum_{k,l}(\mathcal{B}_5)_{i,j,k,l}N_{k,l}$$

$$= \begin{cases} N_{i,i} \text{ if } i \neq j, \\ 0 \text{ if not.} \end{cases}$$

It is the projection of the corresponding line diagonal element.

$$\mathcal{B}_5\tilde{\phantom{.}}N = ((N \odot \mathrm{I})\mathbb{1}\mathbb{1}^{\mathbf{T}}) \odot J$$

$$(\mathcal{B}_6\tilde{\phantom{.}}N)_{i,j} = \sum_{k,l}(\mathcal{B}_6)_{i,j,k,l}N_{k,l}$$

$$= \begin{cases} \sum_{l \neq k} N_{k,l} - \sum_l N_{i,l} - \sum_l N_{l,i} \text{ if } i = j, \\ 0 \text{ if not.} \end{cases}$$

One can recognise $\mathcal{B}_2$ and $\mathcal{B}_3$.

$$\mathcal{B}_6\tilde{\phantom{.}}N = (\mathbb{1}(N \odot J)\mathbb{1}^{\mathbf{T}})\mathrm{I} - \mathcal{B}_2\tilde{\phantom{.}}N - \mathcal{B}_3\tilde{\phantom{.}}N$$

$$(\mathcal{B}_7\tilde{\phantom{.}}N)_{i,j} = \sum_{k,l}(\mathcal{B}_7)_{i,j,k,l}N_{k,l}$$

$$= \begin{cases} \sum_{l \neq i} N_{i,l} - N_{i,j} \text{ if } i \neq j, \\ 0 \text{ if not.} \end{cases}$$

It is just a sum over the line avoiding the element.

$$\mathcal{B}_7\tilde{\phantom{.}}N = (\mathbb{1}\mathbb{1}^{\mathbf{T}}(N \odot J) - N) \odot J$$

$$(\mathcal{B}_8 \tilde{\phantom{}} N)_{i,j} = \sum_{k,l} (\mathcal{B}_8)_{i,j,k,l} N_{k,l}$$

$$= \begin{cases} \sum_l N_{l,i} - N_{j,i} \text{ if } i \neq j, \\ 0 \text{ if not.} \end{cases}$$

It is just a sum over the column corresponding to the line avoiding the transpose element.

$$\mathcal{B}_8 \tilde{\phantom{}} N = ((N \odot J)\mathbb{1}\mathbb{1}^{\mathbf{T}} - N^{\mathbf{T}}) \odot J$$

$$(\mathcal{B}_9 \tilde{\phantom{}} N)_{i,j} = \sum_{k,l} (\mathcal{B}_9)_{i,j,k,l} N_{k,l}$$

$$= \begin{cases} \sum_{l \neq i} N_{j,l} - N_{j,i} \text{ if } i \neq j, \\ 0 \text{ if not.} \end{cases}$$

It is just a sum over the line corresponding to the column avoiding the transpose element.

$$\mathcal{B}_9 \tilde{\phantom{}} N = (\mathbb{1}\mathbb{1}^{\mathbf{T}}(N \odot J) - N^{\mathbf{T}}) \odot J$$

$$(\mathcal{B}_{10} \tilde{\phantom{}} N)_{i,j} = \sum_{k,l} (\mathcal{B}_{10})_{i,j,k,l} N_{k,l}$$

$$= \begin{cases} \sum_l N_{l,j} - N_{i,j} \text{ if } i \neq j, \\ 0 \text{ if not.} \end{cases}$$

It is just a sum over the column avoiding the element.

$$\mathcal{B}_{10} \tilde{\phantom{}} N = ((N \odot J)\mathbb{1}\mathbb{1}^{\mathbf{T}} - N) \odot J$$

$$(\mathcal{B}_{11} \tilde{\phantom{}} N)_{i,j} = \sum_{k,l} (\mathcal{B}_{11})_{i,j,k,l} N_{k,l}$$

$$= \begin{cases} \sum_l N_{l,l} - N_{i,i} - N_{j,j} \text{ if } i \neq j, \\ 0 \text{ if not.} \end{cases}$$

It is just a sum over the diagonal avoiding the two corresponding diagonal elements.

$$\mathcal{B}_{11} \tilde{\phantom{}} N = (\mathbb{1}^{\mathbf{T}}(N \odot \mathrm{I})\mathbb{1})J - \mathcal{B}_3 \tilde{\phantom{}} N - \mathcal{B}_4 \tilde{\phantom{}} N$$

$$(\mathcal{B}_{12} \tilde{\phantom{}} N)_{i,j} = \sum_{k,l} (\mathcal{B}_{12})_{i,j,k,l} N_{k,l}$$

$$= \begin{cases} \sum_l N_{l,l} - N_{i,i} \text{ if } i = j, \\ 0 \text{ if not.} \end{cases}$$

It is just a sum over the diagonal avoiding the corresponding diagonal element.

$$\mathcal{B}_{12} \tilde{\phantom{}} N = (\mathbb{1}^{\mathbf{T}}(N \odot \mathrm{I})\mathbb{1})J - (\mathbb{1}\mathbb{1}^{\mathbf{T}}(N \odot \mathrm{I})) \odot \mathrm{I}$$

$$(\mathcal{B}_{13} \tilde{\phantom{}} N)_{i,j} = \sum_{k,l} (\mathcal{B}_{13})_{i,j,k,l} N_{k,l}$$

$$= \begin{cases} N_{i,j} \text{ if } i \neq j, \\ 0 \text{ if not.} \end{cases}$$

It selects the non-diagonal.

$$\mathcal{B}_{13}\tilde{\cdot}N = N \odot J$$

$$(\mathcal{B}_{14}\tilde{\cdot}N)_{i,j} = \sum_{k,l}(\mathcal{B}_{14})_{i,j,k,l}N_{k,l}$$
$$= \begin{cases} N_{j,i} \text{ if } i \neq j, \\ 0 \text{ if not.} \end{cases}$$

It selects the transpose non-diagonal.

$$\mathcal{B}_{14}\tilde{\cdot}N = N^{\mathbf{T}} \odot J$$

$$(\mathcal{B}_{15}\tilde{\cdot}N)_{i,j} = \sum_{k,l}(\mathcal{B}_{15})_{i,j,k,l}N_{k,l}$$
$$= \begin{cases} \sum_{k \neq l} N_{k,l} - \sum_{i \neq l} N_{i,l} \\ -\sum_{i \neq l} N_{l,i} - \sum_{j \neq l} N_{j,l} \\ -\sum_{j \neq l} N_{l,j} - N_{i,j} - N_{j,i} \text{ if } i \neq j, \\ 0 \text{ if not.} \end{cases}$$

It is in fact a composition of other elements of the basis.

$$\mathcal{B}_{15}\tilde{\cdot}N = (\mathbb{1}^{\mathbf{T}}(N \odot J)\mathbb{1})J - \mathcal{B}_7\tilde{\cdot}N - \mathcal{B}_8\tilde{\cdot}N$$
$$- \mathcal{B}_9\tilde{\cdot}N - \mathcal{B}_{10}\tilde{\cdot}N + \mathcal{B}_{13}\tilde{\cdot}N + \mathcal{B}_{14}\tilde{\cdot}N$$

From all this, we can deduce the following grammar that generates 2-IGN:

$$M \to V_c\mathbb{1}^{\mathbf{T}} \mid M \odot J \mid M \odot \mathrm{I} \mid A$$
$$V_c \to MV_c \mid \mathbb{1}$$

As one can see, there is less operation in the CFG than operators in the basis.

## A.4   GNNs DERIVED FROM DIFFERENT GRAMMARS

This subsection is dedicated to a description of GNNs derived from different grammars of **Q1** experiment (section 4).

Figure 9 depicts a layer of a GNN derived from the exhaustive CFG $G_{\mathcal{L}_3}$. The resulting architecture inherits 3-WL expressive power from theorem 3.1. In Figure 10, a GNN derived from i-$G_{\mathcal{L}_3}$, the CFG obtain during the reduction process of the framework of section 3, is described. Since $^{\mathbf{T}}$ is missing in r-$G(\mathcal{L}_3)$, Figure 11 describes a GNN derived from a grammar containing r-$G(\mathcal{L}_3)$ and $M^{\mathbf{T}}$.

## A.5   $\mathrm{G}^2\mathrm{N}^2$ EXPRESSIVENESS AT FIXED DEPTH

The following proposition ensures that for architectures with at most three layers, the GNN derived from the exhaustive CFG $G(\mathcal{L}_3)$, called E-$\mathrm{G}^2\mathrm{N}^2$, and $\mathrm{G}^2\mathrm{N}^2$ have the same expressive power.

**Proposition A.3** (Graph isomorphism (WL) expressiveness at fixed depth)
*Let $k \in \{1, 2, 3\}$, $G^2N^2$ and E-$G^2N^2$ have the same separative power after $k$ layers.*

*Proof.* Remark that, for scalars $s$ and $s'$, if $ss'$ separates two graphs then $s$ or $s'$ already separate those graphs.

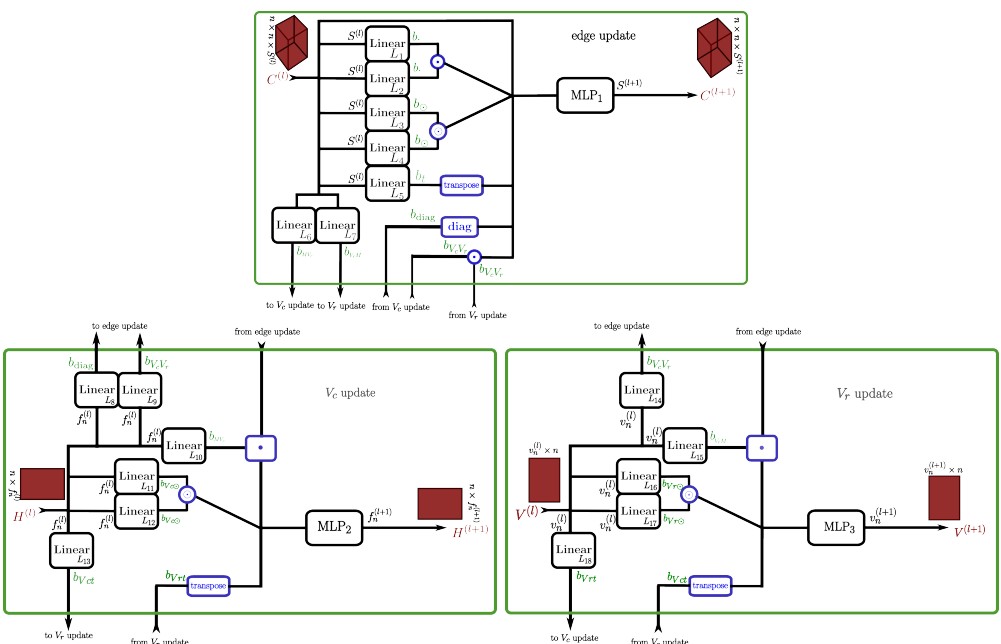

Figure 9: Layer of a GNN derived from $G_{\mathcal{L}_3}$.

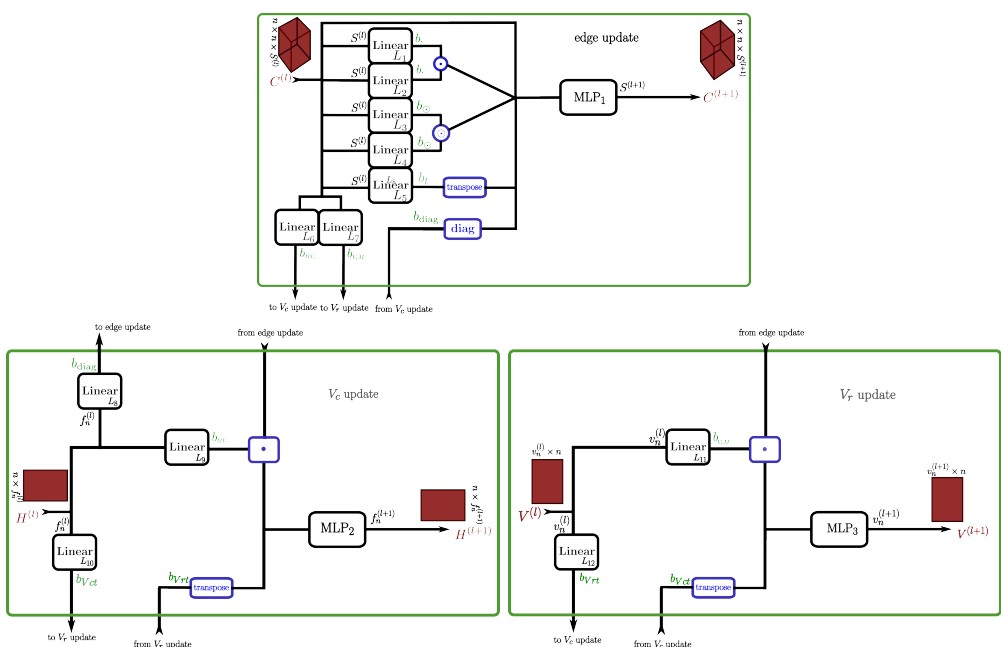

Figure 10: Layer of a GNN derived from i-$G_{\mathcal{L}_3}$.

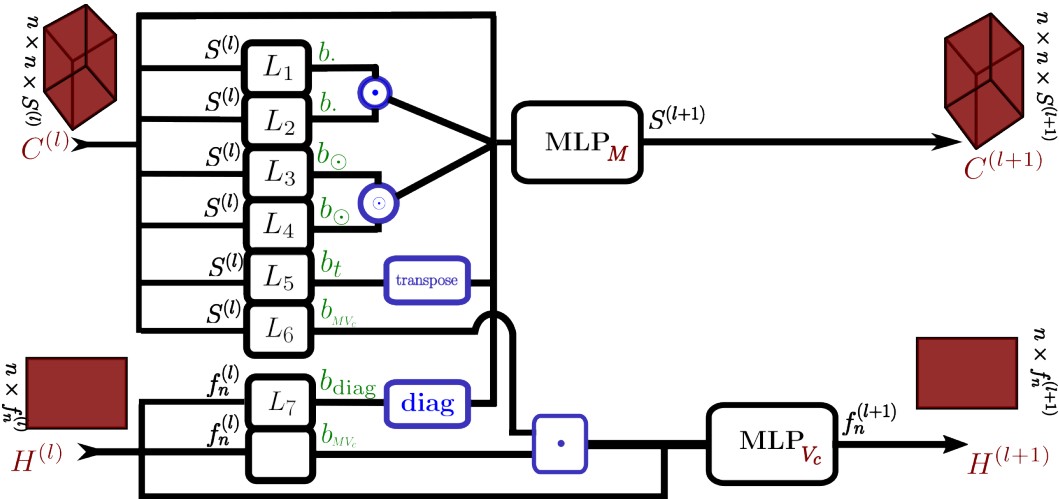

Figure 11: Layer of a GNN derived r-$G_{\mathcal{L}_3} \cup \{M^{\mathbf{T}}\}$.

We denote by $M_G^{(k)}$, $V_{cG}^{(k)}$ and $S_G^{(k)}$, respectively the set of matrices, vectors and scalars that G$^2$N$^2$ can compute at layer $k$. We also denotes by $M_E^{(k)}$, $V_{cE}^{(k)}$, $V_{rE}^{(k)}$ and $S_E^{(k)}$, respectively the set of matrices, column and row vectors and scalars that E-G$^2$N$^2$ can compute at layer $k$. Remark that thanks to the skip connection in both layer, these sets are increasing for inclusion with respect to $k$. For example $S_G^{(k)} \subset S_G^{(k+1)}$.

We will show that $S_G^{(k)} = S_E^{(k)}$ for $k \in \{1, 2, 3\}$.

First of all we have that $M_E^{(0)} = M_G^{(0)} = \{A\}$ and $V_{cE}^{(0)} = V_{rE}^{(0)} = V_{cG}^{(0)} = \{\mathbb{1}\}$ by construction of the architectures.

Let compare two architectures with only one layer, we have that $V_{cE}^{(1)} = V_{rE}^{(1)} = V_{cG}^{(1)}$ since $\mathbb{1} \odot \mathbb{1} = \mathbb{1}$, but $M_E^{(1)} = M_G^{(1)} \cup \{vw^T, v, w \in V_{cG}^{(0)}\}$. Since $\{vw^T, v, w \in V_{cG}^{(0)}\} = \mathbb{1}\mathbb{1}^{\mathbf{T}}$ and this matrix can be approximated by the bias of any of the matrix linear of G$^2$N$^2$, we have $M_E^{(1)} = M_G^{(1)}$. Thus $S_E^{(1)} = S_G^{(1)}$.

Let compare two architectures with two layers, we have that $V_{cE}^{(2)} = V_{rE}^{(2)} = V_{cG}^{(2)}$ since like in Maron et al. (2019a) the vector Hadamard product can be approximated with a multi layer perceptron, but $M_E^{(2)} = M_G^{(2)} \cup \{vw^T, v, w \in V_{cG}^{(1)}\}$. Let $N = vw^T$, $v, w \in V_{cG}^{(1)}\}$, then applying a readout function on $N$ would result in

$$1^T N 1 = \underbrace{1^T v}_{\in S_G^{(1)}} \underbrace{w^T 1}_{\in S_G^{(1)}}.$$

From that we have that $S_E^{(2)} = S_G^{(2)}$ since the final decision multilayer perceptron can approximate $ss'$ for any scalar $s$, $s'$. So our hypothesise is true for $k = 2$.

Assume that we compare architectures with $k = 3$ layers.

Since $M_E^{(2)} = M_G^{(2)} \cup \{vw^T, v, w \in V_{cG}^{(1)}\}$, we have for the matrix case,

$$
\begin{aligned}
M_E^{(3)} = \; & M_G^{(3)} \cup \{vw^T, v, w \in V_{cG}^{(2)}\} \\
& \cup \{M(vw^T) \text{ or } (vw^T)M, M \in M_E^{(2)}, v, w \in V_{cG}^{(1)}\} \\
& \cup \{M \odot (vw^T), M \in M_E^{(2)}, v, w \in V_{cG}^{(1)}\}.
\end{aligned}
$$

We have three cases here. For $N \in \{vw^T, v, w \in V_{cG}^{(2)}\}$, it is the same situation than for $k = 2$.

For $N = M(vw^T)$, $M \in M_E^{(2)}$, $v, w \in V_{cG}^{(1)}$, either $M \in M_G^{(2)}$ or $M \in \{vw^T, v, w \in V_{cG}^{(1)}\}$. In the first case $Mv \in V_{cG}^{(3)}$ and since $V_{cG}^{(1)} \subset V_{cG}^{(3)}$, we are in the same case than for $k = 2$. In the second case, there exists $v'$ and $w' \in V_{cG}^{(1)} \subset V_{cG}^{(2)}$ such that $M = v'(w')^T$, which means that after the readout function, we have

$$1^T N 1 = \underbrace{1^T v'}_{\in S_G^{(2)}} \underbrace{(w')^T v}_{\in S_G^{(3)}} \underbrace{w^T 1}_{\in S_G^{(2)}}.$$

For $N = M \odot (vw^T)$, $M \in M_E^{(2)}$, $v, w \in V_{cG}^{(1)}$, either $M \in M_G^{(2)}$ or $M \in \{vw^T, v, w \in V_{cG}^{(1)}\}$. In the first case, we have that $1^T(M \odot (vw^T))1 \in S_G^{(3)}$. Indeed,

$$1^T(M \odot (vw^T))1 = v^T M w = 1^T \underbrace{(v \odot Mw)}_{\in V_{cG}^{(3)}}$$

We have for the vector case,

$$V_{cE}^{(3)} = V_{rE}^{(3)} = V_{cG}^{(3)} \cup \{vw^T v', v, w \in V_{cG}^{(1)}, v' \in V_{cG}^{(2)}\}.$$

Since $w^T v' \in S_G^{(3)}$ for all $w, v' \in V_{cG}^{(2)}$, we have that $S_E^{(3)} = S_G^{(3)}$.

$\square$

## B Spectral response of $\text{ML}(\mathcal{L}_3)$

The graph Laplacian is the matrix $L = D - A$ (or $L = \mathrm{I} - D^{-\frac{1}{2}} A D^{-\frac{1}{2}}$ for the normalised Laplacian) where $D$ is the diagonal degree matrix. Since $L$ is positive semidefinite, its eigendecomposition is $L = U \text{diag}(\lambda) U^{\mathbf{T}}$ with $U \in \mathbb{R}^{n \times n}$ orthogonal and $\lambda \in R_+^n$. By analogy with the convolution theorem, one can define graph filtering in the frequency domain by $\tilde{x} = U \text{diag}(\Omega(\lambda)) U^{\mathbf{T}} x$ where $\Omega$ is the filter applied in the spectral domain.

**Lemme B.1**
*Given $A$ the adjacency matrix of a graph, $\text{ML}(\mathcal{L}_3)$ can compute the graph Laplacian $L$ and the normalised Laplacian $L_n$ of this graph.*

*Proof.* $\text{ML}(\mathcal{L}_3)$ can produce $A^2 \odot \mathrm{I}$ which is equal to $D$. Thus it can compute $L = D - A$. For the normalised Laplacian, since the point-wise application of a function does not improve the expressive power of $\text{ML}(\mathcal{L}_3)$ (Geerts (2020)), $D^{-\frac{1}{2}}$ is reachable by $\text{ML}(\mathcal{L}_3)$. Thus, the normalised Laplacian $D^{-\frac{1}{2}} L D^{-\frac{1}{2}}$ can be computed. $\square$

As in Balcilar et al. (2020), we define the spectral response $\phi \in \mathbb{R}^n$ of $C \in \mathbb{R}^{n \times n}$ as $\phi(\lambda) = \text{diagonal}(U^{\mathbf{T}} C U)$ where diagonal extracts the diagonal of a given square matrix. Using spectral response, Balcilar et al. (2020) shows that most existing MPNNs act as low-pass filters while high-pass and band-pass filters are experimentally proved to be necessary to increase model expressive power.

**Theorem B.2**
*For any continuous filter $\Omega$ in the spectral domain of the normalised Laplacian, there exists a matrix in $\text{ML}(\mathcal{L}_3)$ such that its spectral response approximate $\Omega$.*

*Proof.* The spectrum of the normalised Laplacian is included in $[0, 2]$, which is compact. Thanks to Stone-Weierstraß theorem, any continuous function can be approximated by a

polynomial function. We just have to ensure the existence of a matrix in $\mathrm{ML}(\mathcal{L}_3)$ such that its spectral response is a polynomial function.

For $k \in \mathbb{N}$, the spectral response of $L^k$ is $\lambda^k$ since we have

$$U^{\mathbf{T}} L^k U = U^{\mathbf{T}} (U \operatorname{diag}(\lambda) U^{\mathbf{T}})^k U$$
$$= U^{\mathbf{T}} U \operatorname{diag}(\lambda)^k U^{\mathbf{T}} U = \operatorname{diag}(\lambda)^k$$

From Lemma B.1, $\mathrm{ML}(\mathcal{L}_3)$ can compute $L$, and thus it can compute $L^k$ for any $k \in \mathbb{N}$. Since $\mathrm{ML}(\mathcal{L}_3)$ can produce all the matrices with a monome spectral response and since the function that gives the spectral response to a given matrix is linear, $\mathrm{ML}(\mathcal{L}_3)$ can produce any matrices with a polynomial spectral response. □

This section shows that a 3-WL GNN should be able to approximate any type of filter.

## C  EXPERIMENTS

### C.1  EXPERIMENTAL SETTING

In the experiments, all the linear blocks of a layer are set at the same width $S^{(l)} = b_{\otimes}^{(l)} = b_{\odot}^{(l)} = b_{\mathrm{diag}}^{(l)}$. This means that $\mathrm{MLP}_M^{(l)}$ takes as input a third order tensor of dimensions $n \times n \times 4S^{(l)}$ and $\mathrm{MLP}_{V_c}^{(l)}$ takes as input a matrix of dimensions $n \times 2S^{(l)}$. At each layer, the MLP depth is always 2 and the intermediate layer doubled the input dimension.

### C.2  QM9

For this experiment, there are 4 edge attributes and 11 node features. We use 3 layers with $S^{(l)} = f_n^{(l)} = 64$ when learning one target at a time and $S^{(l)} = f_n^{(l)} = 32$ in the other experiment for $l \in \{1, 2, 3\}$. The vector readout function is a sum over the components of $H^{(3)}$ and the matrix readout function is a sum over the components of the diagonal and the off-diagonal parts of $\mathcal{C}^{(3)}$. Finally, 3 fully connected layers, with respective dimension$(512/256/(1 \text{ or } 12))$ are applied before using an absolute error loss. Complete results on this dataset can be found in Table 4.

Table 4: Results on QM9 dataset predicting each target at a time. The metric is MAE, the lower, the better.

| Target | 1-GNN | 1-2-3-GNN | DTNN | Deep LRP | NGNN | I$^2$-GNN | PPGN | G$^2$N$^2$ |
|--------|-------|-----------|------|----------|------|-----------|------|-----------|
| $\mu$ | 0.493 | 0.476 | 0.244 | 0.364 | 0.428 | 0.428 | 0.0934 | **0.0703** |
| $\alpha$ | 0.78 | 0.27 | 0.95 | 0.298 | 0.29 | 0.230 | 0.318 | **0.127** |
| $\epsilon_{\mathrm{homo}}$ | 0.00321 | 0.00337 | 0.00388 | 0.00254 | 0.00265 | 0.00261 | 0.00174 | **0.00172** |
| $\epsilon_{\mathrm{lumo}}$ | 0.00355 | 0.00351 | 0.00512 | 0.00277 | 0.00297 | 0.00267 | 0.0021 | **0.00153** |
| $\Delta\epsilon$ | 0.0049 | 0.0048 | 0.0112 | 0.00353 | 0.0038 | 0.0038 | 0.0029 | **0.00253** |
| $R^2$ | 34.1 | 22.9 | 17.0 | 19.3 | 20.5 | 18.64 | 3.78 | **0.342** |
| ZPVE | 0.00124 | 0.00019 | 0.00172 | 0.00055 | 0.0002 | 0.00014 | 0.000399 | **0.0000951** |
| $U_0$ | 2.32 | 0.0427 | 2.43 | 0.413 | 0.295 | 0.211 | 0.022 | **0.0169** |
| $U$ | 2.08 | 0.111 | 2.43 | 0.413 | 0.361 | 0.206 | 0.0504 | **0.0162** |
| $H$ | 2.23 | 0.0419 | 2.43 | 0.413 | 0.305 | 0.269 | 0.0294 | **0.0176** |
| $G$ | 1.94 | 0.0469 | 2.43 | 0.413 | 0.489 | 0.261 | 0.024 | **0.0214** |
| $C_v$ | 0.27 | 0.0944 | 2.43 | 0.129 | 0.174 | 0.0730 | 0.144 | **0.0429** |

### C.3  TUD

The parameter setting for each of the 6 experiments related to this dataset can be found in Table 5. Complete results on this dataset are given in Table 6.

Table 5: G$^2$N$^2$ parameters detail for each dataset in our experiments on TU

| parameters | MUTAG | PTC | Proteins | NCI1 | IMDB-B | IMDB-M |
|---|---|---|---|---|---|---|
| node features | 7 | 22 | 3 | 37 | 1 | 1 |
| edge features | 6 | 0 | 0 | 0 | 0 | 0 |
| # of G$^2$N$^2$ layer $=l_m$ | 3 | 3 | 3 | 3 | 3 | 3 |
| $f_n^{(l)}$ $l \in [\![1,l_m]\!]$ | 16 | 32 | 32 | 64 | 32 | 32 |
| $S^{(l)}$ $l \in [\![1,l_m]\!]$ | 16 | 32 | 32 | 64 | 32 | 32 |
| readout dimension | 256/128/1 | 512/256/1 | 512/256/1 | 128/64/1 | 512/256/1 | 512/256/3 |
| loss | BCEloss | BCEloss | BCEloss | BCEloss | BCEloss | CEloss |

Table 6: Results on TUD dataset. The metric is accuracy, the higher, the better.

| Dataset | MUTAG | PTC | Proteins | NCI1 | IMDB-B | IMDB-M |
|---|---|---|---|---|---|---|
| WL kernel (Shervashidze et al. (2011)) | 90.4±5.7 | 59.9±4.3 | 75.0±3.1 | **86.0±1.8** | 73.8±3.9 | 50.9±3.8 |
| GNTK (Du et al. (2019)) | 90.0±8.5 | 67.9±6.9 | 75.6±4.2 | 84.2±1.5 | 76.9±3.6 | 52.8±4.6 |
| DGCNN (Zhang et al. (2018)) | 85.8±1.8 | 58.6±2.5 | 75.5±0.9 | 74.4±0.5 | 70.0±0.9 | 47.8±0.9 |
| IGN (Maron et al. (2019b)) | 83.9±13.0 | 58.5±6.9 | 76.6±5.5 | 74.3±2.7 | 72.0±5.5 | 48.7±3.4 |
| GIN (Xu et al. (2019)) | 89.4±5.6 | 64.6±7.0 | 76.2±2.8 | 82.7±1.7 | 75.1±5.1 | 52.3±2.8 |
| PPGNs (Maron et al. (2019a)) | 90.6±8.7 | 66.2±6.6 | 77.2±4.7 | 83.2±1.1 | 73.0±5.8 | 50.5±3.6 |
| Natural GN (de Haan et al. (2020)) | 89.4±1.60 | 66.8±1.79 | 71.7±1.04 | 82.7±1.35 | 74.8±2.01 | 51.3±1.50 |
| WEGL (Kolouri et al. (2020)) | N/A | 67.5±7.7 | 76.5±4.2 | N/A | 75.4±5.0 | 52.3±2.9 |
| GIN+GraphNorm (Cai et al. (2021)) | 91.6±6.5 | 64.9±7.5 | 77.4±4.9 | 82.7±1.7 | 76.0±3.7 | N/A |
| GSNs (Bouritsas et al. (2022)) | 92.2±7.5 | 68.2±7.2 | 76.6±5.0 | 83.5±2.0 | **77.8±3.3** | **54.3±3.3** |
| G$^2$N$^2$ | **92.5±4.3** | **72.3±6.3** | **80.1±3.7** | 82.8±0.9 | 76.8±2.8 | 54.0±2.9 |

## C.4 SPECTRAL DATASET

This dataset is composed of three 2D grids of size 30x30, for respectively training, validation, and testing. We use 3 layers of G$^2$N$^2$ with $S^{(l)} = 32$ and $f_n^{(l)} = 32$ for $l \in \{1,2,3\}$. Our readout function is the identity over the last node embedding and a sum over the line of the last edge embedding. We finally apply two fully connected layers on the output of the readout function and then use Mean Square Error (MSE) loss to compare the output to the ground truth.

