# OpenReview forum: "G$^2$N$^2$ : Weisfeiler and Lehman go grammatical"
_ICLR.cc/2024/Conference — ICLR 2024 poster_

### Official Review · Reviewer_giVQ · 2023-10-18

**Soundness:** 4 excellent
**Presentation:** 4 excellent
**Contribution:** 3 good
**Rating:** 8
**Confidence:** 3

**Summary:**

The paper proposes a new architecture for GNNs that captures precisely the expressive power of 3WL. This architecture is based on a grammatical representation of a language over graphs that has the same expressive power as 3WL. The idea is that this new architecture permits a more efficient implementation than the 3WL-based GNNs, which are known not to scale well in practical scenarios.

**Strengths:**

- The paper is very polished and easy to follow
- The topic is timely and the problem practically relevant
- Experiments confirm the suitability if the approach

**Weaknesses:**

There is only one criticism I make to the paper and it is the lack of search for a principled explanation of why the GNNs based on MATLANG are more efficient than the ones based on 3WL.

**Questions:**

Could you please comment further on the main criticism I posed above: what do you think is the main reason the MATLANG-based GNNs are more practically suitable than the standard ones based on 3WL? This came as a big surprise to me, and it is a bit dissatisfying to stay with an explanation based on experiments only. I feel that a more principled, perhaps theoretical explanation, is lacking.

---

> ### Author Response · Authors · 2023-11-17
> **Global reply to the Reviewer giVQ**
>
> We thank the reviewer for his comments and question. We are very pleased that he appreciates the writing of our paper, its relevance and the suitability of our approach. Below, we reply to his question, hoping that these responses will clarify the positioning of MATLANG-based GNNs in the 3-WL literature.

---

> ### Author Response · Authors · 2023-11-17
> **Reply to the question about the reason why MATLANG-based GNNs are more practically suitable that the standard ones based on 3WL**
>
> *Reviewer question : what do you think is the main reason the MATLANG-based GNNs are more practically suitable than the standard ones based on 3WL? This came as a big surprise to me, and it is a bit dissatisfying to stay with an explanation based on experiments only. I feel that a more principled, perhaps theoretical explanation, is lacking.*
>
> In the experiments of section 4, PPGN is the only model to have a provable 3-WL expressive power Maron et al. (2019). Indeed, if 3-GNN has a 3-WL expressive power, its relaxed version 1-2-3-GNN does not provably reach this expressiveness.
>
> In the subsection 3.5 of the paper, we present some theoretical explanation concerning the superiority of G$^2$N$^2$ on PPGN for downstream tasks. These explanations take their root in the CFG we build from the PPGN model (please see the appendix A.3). We argue that the CFG of PPGN misses a variable ($V_c$) and a lot of rules of r-$G(L_3)$.
>
> During the rebuttal period, thanks to the questions of the other reviewers, we investigate the expressive power of our architecture at a fixed number of layer. This investigation leads to the following proposition, which we add to appendix A.5.
>
> **Proposition[Graph isomorphism (WL) expressiveness at fixed depth]**
>
> *Let $k \in \{1,2,3 \}$, G$^2$N$^2$ and E-G$^2$N$^2$ have the same separative power after $k$ layers.*
>
>
> The proof of this proposition, also available in appendix A.5, has highlighted the benefit of the $V_c$ variable and in particular of the rule $MV_c$. Since PPGN needs more layers to compute a matrix carrying this rule its expressive power at a fixed number of layer is lower.
>
> Another important aspect is that PPGN needs to approximate some of these missing rules using MLPs. To guarantee such  an approximation, a certain width and depth for MLP are needed. G$^2$N$^2$ does not suffer from these computational constraints since it only needs to provide linear combinations as arguments of the operations.
>
> H. Maron, H. Ben-Hamu, H. Serviansky, and Y. Lipman. Provably powerful graph networks.
> Advances in neural information processing systems, 32, 2019.

---

### Official Review · Reviewer_3x4U · 2023-11-01

**Soundness:** 4 excellent
**Presentation:** 4 excellent
**Contribution:** 4 excellent
**Rating:** 6
**Confidence:** 2

**Summary:**

This paper presents a framework to convert context-free rules over an algebraic matrix language into a GNN architecture. Using this framework, they produce a WL-3 GNN as follows: (1) they write down a set of context-free rules producing a language that is just as expressive as 3-WL, (2) they reduce this set of rules into a smaller set of rules, and (3) they translate these rules directly into a GNN architecture. The resulting architecture performs competitively in practice, outperforming various existing GNNs on a variety of tasks.

**Strengths:**

(1) While this is not my area, the contribution of the paper seems strong in that it presents a framework for designing GNN architectures that implement a given CFG.

(2) The experiments seem strong, and the proposed GNN is both provably expressive and performs competitively compared to existing architectures.

(3) The paper is well-written, clear, and well-organized.

**Weaknesses:**

Some minor weaknesses are discussed in the questions section.

**Questions:**

(1) While many CFGs are equally expressive if we can apply their rules an arbitrary number of times, it seems like what we actually care about is the expressiveness of the grammar after L rule applications, given that in practice our GNNs are finite depth. In light of this intuition, the paper might benefit from some discussion of which CFGs are preferable, given that they have the same expressive power.

(2) From my understanding, it seems that this architecture outperforms other architectures in some datasets but not others (table 6), but has the advantage of being the strongest GNN out of those with provable expressivity (since it dominates PPGN). Is this understanding correct? If so, what is the utility of having provable expressivity, beyond the model's performance on the datasets?

---

> ### Author Response · Authors · 2023-11-17
> **Global reply to the Reviewer 3x4U**
>
> We thank the reviewer for his constructive comments and questions. We are pleased that he appreciates the writing of our paper and that he considers our contributions and experiments strong. Below, we reply to his two questions in two separated comments, hoping that these responses will clarify some aspects of our work.

---

> ### Author Response · Authors · 2023-11-17
> **Reply to the question about the impact of the number of GNN layers on the expressiveness**
>
> *Reviewer question : While many CFGs are equally expressive if we can apply their rules an arbitrary number of times, it seems like what we actually care about is the expressiveness of the grammar after L rule applications, given that in practice our GNNs are finite depth. In light of this intuition, the paper might benefit from some discussion of which CFGs are preferable, given that they have the same expressive power.*
>
> The reviewer is right, many CFGs share the same expressive power. As shown by Theorem 3.2. of the paper, it is the case for r-G(L3) and G(L3). We would like to emphasize that this theoretical result assumes an arbitrary depth, as usually done in the literature when considering the expressive power of a GNN model (Zhang et al. (2023)).
>
> Despite this statement, it is true that the number of rule applications (i.e. the number of layers in the GNN) has an impact on the expressiveness. Thus, questioning the differences between the different CFGs and the impact of their depth is very relevant. Please note that this question has also been raised by reviewer HFhT.
>
> We have examined this aspect from a theoretical point of view during the rebuttal period. In this limited time, we manage to provide the following proposition showing that both models have the same separative power for a fixed number of layers $k$ such that $k \leq 3$.
>
> **Proposition[Graph isomorphism (WL) expressiveness at fixed depth]**
>
> *Let $k \in \{1,2,3 \}$, G$^2$N$^2$ and E-G$^2$N$^2$ have the same separative power after $k$ layers.*
>
>
> This result is mentioned in Section 4, question **Q1**. The proposition and its full proof have been added in appendix A.5. The rationale behind the proof is that graph isomorphism (i.e. the WL hierarchy) measures expressiveness at graph-level, as mentioned in section 2 of our paper and in Zhang et al. (2023). As a consequence, for measuring the expressive power at a given depth, we are interested in strings that compute a scalar.
>
> Generalizing this result to deeper models, with $k>3$, needs further investigations that will be led in our future work.
> We thank the reviewer again for this question which led us to find this new proposition which theoretically strengthens our paper since all our experimental results are obtained with $k=3$.
>
> B. Zhang, C. Fan, S. Liu, K. Huang, X. Zhao, J. Huang, and Z. Liu. The expressive power
> of graph neural networks: A survey. arXiv preprint arXiv:2308.08235, 2023.

---

> ### Author Response · Authors · 2023-11-17
> **Reply to the question on the results of G$^2$N$^2$ on some datasets**
>
> *Reviewer question : From my understanding, it seems that this architecture outperforms other architectures in some datasets but not others (table 6), but has the advantage of being the strongest GNN out of those with provable expressivity (since it dominates PPGN). Is this understanding correct? If so, what is the utility of having provable expressivity, beyond the model's performance on the datasets?*
>
> It is true that although G$^2$N$^2$ has a provable expressiveness, it does not outperform all other architectures for all the datasets in Table 6. Our model is slightly outperformed by some architectures on the IMDB and NCI1 datasets. [Please note that during the rebuttal we realised that there was an error in table 6, as G$^2$N$^2$ is not ranked second but third on the IMDB-B dataset. We apologise for this typo which has been corrected in the new version].
>
> In the case of the IMDB datasets and the NCI1 dataset, our model is outperformed by the GSN model, proposed in Bouritsas et al. (2022). This is due to the specific characteristics of these datasets since graphs in IMDB consist of a union of dense subgraphs (cliques). Hence, GSN is based on feature augmentation with subgraph counting both at edge- and node-level. More precisely, their best model on IMDB computes 5-clique at edge-level, which is not computable by 3-WL GNNs. In the same vein, for the NCI1 dataset, they compute the 15-cycle at edge level when achieving their best results, which again is not computable by 3-WL GNNs.
>
> Such results illustrate the importance of the expressive power of models on the downstream tasks performance. They also highlight that the WL hierarchy is not the only way to assess expressive power, since substructure counting abilities also impacts these performance. This aspect has been highlighted in many recent papers such as Frasca et al. (2022); Huang et al.
> (2023). These considerations open the door to the building of grammars based on subgraph counting. We plan to investigate this aspect in our future works.
>
> Concerning the performance of GNTK on IMDB which are comparable with ours, as mentioned in Chami et al. (2022), GNTK is a graph Kernel based method. One can conjectures that the kernel used by GNTK well fit on the specificity of IMDB dataset. Further investigations are needed to bridge the gap between kernel based methods and expressiveness issues.
>
> G. Bouritsas, F. Frasca, S. Zafeiriou, and M. M. Bronstein. Improving graph neural network
> expressivity via subgraph isomorphism counting. IEEE Transactions on Pattern Analysis
> and Machine Intelligence, 45(1):657–668, 2022.
>
> I. Chami, S. Abu-El-Haija, B. Perozzi, C. R´e, and K. Murphy. Machine learning on graphs:
> A model and comprehensive taxonomy. The Journal of Machine Learning Research, 23(1):
> 3840–3903, 2022.
>
> F. Frasca, B. Bevilacqua, M. M. Bronstein, and H. Maron. Understanding and extending
> subgraph gnns by rethinking their symmetries. In Advances in Neural Information
> Processing Systems, 2022.
>
> Y. Huang, X. Peng, J. Ma, and M. Zhang. Boosting the cycle counting power of graph
> neural networks with I2
> -GNNs. In The Eleventh International Conference on Learning
> Representations, 2023.

---

> > ### Comment · Reviewer_3x4U · 2023-11-23
> >
> > Thanks for your comprehensive answers.

---

### Official Review · Reviewer_HFhT · 2023-11-01

**Soundness:** 3 good
**Presentation:** 3 good
**Contribution:** 3 good
**Rating:** 6
**Confidence:** 4

**Summary:**

This paper investigates the expressive power of 3-WL from the aspect of formal language. The authors show that 3-WL is equivalent to a context-free grammar (CFG), and propose a reduced CFG that preserves the same expressiveness. Based on the reduced CFG, the authors develop a new WL algorithm and GNN model that match the expressiveness of 3-WL. The new GNN model achieves competitive performance and efficiency on downstream tasks.

**Strengths:**

There are some positive points of this paper.
* The paper is well-written and easy to follow.
* The paper exploits the formal language equivalence to investigate the GNN model and design a new GNN model, which I think is a promising direction for future research.

**Weaknesses:**

I have some concerns about the paper as follows:
* I am not convinced by the novelty and the contribution of the paper, as the CFG $G_\mathcal{L_3}$​​ seems to be a straightforward derivation of the MATLANG.
* The validation and discussion of the reduced CFG and the corresponding GNN may be insufficient, both empirically and theoretically. I have some questions for the authors below.

**Questions:**

* In the theoretical aspect, although the two CFGs r-$G_{\mathcal{L}_3}$ and $G_{\mathcal{L}_3}$ have the same expressive power, it may take more steps for r-$G_{\mathcal{L}_3}$ than $G_{\mathcal{L}_3}$ to generate the same string. Therefore, how can $G^2N^2$ match the expressiveness of the ordinary 3-WL GNN with a fixed number of layers?
* I would also like to see how the new GNN model performs on the ZINC-12k and ZINC-full datasets, which are widely used benchmarks for molecular property prediction.

---

> ### Author Response · Authors · 2023-11-17
> **Global reply to the Reviewer HFhT:**
>
> We thank the reviewer for his constructive comments and questions. We are pleased that he appreciates the writing of our paper and that he thinks our contributions offer promising directions for future research. Below, we respond to his two questions in two separated comments, hoping that these responses will also address the weaknesses he identifies.

---

> ### Author Response · Authors · 2023-11-17
> **Reply to the question about the impact of the grammar reduction and the number of GNN layers on the expressiveness**
>
> *Reviewer question : In the theoretical aspect, although the two CFGs $r-G(L3)$ and $G(L3)$ have the same expressive power, it may take more steps for $r-G(L3)$ than $G(L3)$ to generate the same string.  Therefore, how can $G^2N^2$ match the expressiveness of the ordinary 3-WL GNN with a fixed number of layers ?*
>
> The reviewer is right : r-G(L3) and G(L3) have the same expressive power, as shown by theorem 3.2. of the paper. It is also true that r-G(L3) is not able to generate all the strings that can be generated by G(L3) since more operations are available in G(L3). So, questioning the necessity of using more GNN layers with G$^2$N$^2$ than with the GNN built from G(L3) (called E-G$^2$N$^2$ in this reply) is very relevant.
>
> We have examined this aspect from a theoretical point of view during the rebuttal period. In this limited time, we manage to provide the following proposition showing that both models have the same separative power for a fixed number of layers $k$ such that $k \leq 3$.
>
> **Proposition[Graph isomorphism (WL) expressiveness at fixed depth]**
>
> *Let $k \in \{1,2,3 \}$, G$^2$N$^2$ and E-G$^2$N$^2$ have the same separative power after $k$ layers.*
>
>
> This result is mentioned in Section 4, question **Q1**. The proposition and its full proof have been added in appendix A.5. The rationale behind the proof is that graph isomorphism (i.e. the WL hierarchy) measures expressiveness at graph-level, as mentioned in section 2 of our paper and in Zhang et al. (2023). As a consequence, for measuring the expressive power at a given depth, we are interested in strings that compute a scalar.
>
> Generalizing this result to deeper models, with $k>3$, needs further investigations that will be led in our future work.
>
> We thank the reviewer again for this question which led us to find this new proposition which theoretically strengthens our paper since all our experimental results are obtained with $k=3$.
>
> B. Zhang, C. Fan, S. Liu, K. Huang, X. Zhao, J. Huang, and Z. Liu. The expressive power
> of graph neural networks: A survey. arXiv preprint arXiv:2308.08235, 2023.

---

> > ### Comment · Reviewer_HFhT · 2023-11-19
> >
> > Thanks for your reply. I think this point is compelling. It would be nice to see this incorporated into the revision. I will raise my score to a 6 regardless of the results of the experiments.

---

> ### Author Response · Authors · 2023-11-17
> **Reply to the question about the performance of G$^2$N$^2$ on ZINC datasets**
>
> *Reviewer question : I would also like to see how the new GNN model performs on the ZINC-12k and ZINC-full datasets, which are widely used benchmarks for molecular property prediction.*
>
> We have launched some experiments on these two datasets. The results will be provided and discussed in a new comment as soon as they are available.

---

> > ### Author Response · Authors · 2023-11-21
> > **Reply to the question about the performance of G$^2$N$^2$ on ZINC datasets**
> >
> > We would like to thank the reviewer for his feedback on our answer to his first question. We are very pleased that he found it compelling.
> >
> > Dealing with the experiments on the ZINK-12k and ZINK-full datasets, we only compare G$^2$N$^2$ with PPGN Maron et al. (2019))  due to time restriction. For this comparison, we follow the training procedure of  Dwivedi et al. (2020). As the training protocol requires to keep the number of parameters under 500 k, we take this limit as target when designing the models. In this context, we apply 6 layers of G$^2$N$^2$ with an embedding size of 32 for both edges and nodes and 7 layers of PPGN with an embedding size of 64. In both cases, we apply a decision layer of size 128/64/1. The results of these experiments are reported in the following table.
> >
> >
> > Results on ZINK, MAE the lower the better
> >
> > | Model  | ZINK-12k | ZINK-FULL  |
> > |:---:|:---:|:---:|
> > | G$^2$N$^2$  | $0.077 \pm 0.004$ | $0.023 \pm 0.002$ |
> > | PPGN | $0.082 \pm 0.003$ | $0.038 \pm 0.002$ |
> >
> >
> >
> > These results should be taken with caution as we did not have enough time to optimize the hyperparameters of both models. Nevertheless, they show that G$^2$N$^2$, even under-optimized, achieves SOTA performance on these datasets, as shown by the Table 6 of the paper (Kong et al. (2023)) accepted at NEURIPS 2023.
> >
> > We will conduct further experiments as we are convinced that G$^2$N$^2$ can achieve better results, particularly on ZINK-FULL.
> >
> > Since these results are incomplete, our proposal to the reviewer is to provide them, as well as the source code used for producing them, on our github repository. Other suggestions by the reviewer are welcome.
> >
> > V. P. Dwivedi, C. K. Joshi, T. Laurent, Y. Bengio, and X. Bresson. Benchmarking graph
> > neural networks. arXiv preprint arXiv:2003.00982, 2020
> >
> > L. Kong, J. Feng, H. Liu, D. Tao, Y. Chen, and M. Zhang. Mag-gnn: Reinforcement learning
> > boosted graph neural network. arXiv preprint arXiv:2310.19142, 2023.
> >
> > H. Maron, H. Ben-Hamu, H. Serviansky, and Y. Lipman. Provably powerful graph networks.
> > Advances in neural information processing systems, 32, 2019

---

### Meta-Review · Area_Chair_Sc7W · 2023-12-06

**Metareview:**

This paper introduces a framework for designing expressive GNNs by using context-free-language formed by basic linear computations of the graph adjacency matrix. This approach can obtain GNNs that match the 3-WL test while being slightly simpler than prior work like PPGN. All reviewers are positive regarding the novelty and contribution of this paper. I thus recommend acceptance following the majority.

**Justification For Why Not Higher Score:**

This paper has limited accessibility. Some reviewers assigned before could not fully understand the submission, and the impact of this work might be limited.

**Justification For Why Not Lower Score:**

All reviewers are positive regarding the novelty and contribution of this paper.

---

### Decision · Program_Chairs · 2024-01-16

Accept (poster)